# REVISITING HYPERNETWORK IN MODEL-HETEROGENEOUS PERSONALIZED FEDERATED LEARNING

## ABSTRACT

Most recent personalized federated learning research focuses on heterogeneous models among clients. However, the current methods rely on external data, model decoupling, and partial learning, which makes them sensitive to the settings. In contrast, we revisit hypernetworks and leverage their strong generalization ability to propose the first practical method for personalized federated learning. We first propose a **m**odel-**h**eterogeneous **p**ersonalized **fed**erated learning framework based on **hypern**etworks, **MH-pFedHN**, which quantifies clients with different architectures and generates client-specific model parameters using our designed customized embedding vectors through a server-side hypernetwork. Besides a feature extractor, our hypernetwork consists of multiple heads, where clients with similar parameter sizes use the same number of customized embedding vectors and share the same head. Thus, our MH-pFedHN enables knowledge sharing across different architectures and reduces the computation of parameter generation. To push the performance limits, we introduce a `plug-in` component, a lightweight yet effective global model, to enhance learning and generalization capabilities, named **MH-pFedHNGD**. Our framework does not utilize external data and does not require the disclosure of client model architectures, thereby effectively ensuring security and demonstrating great potential. Experiments across various models and tasks demonstrate that our approach outperforms standard baselines and exhibits strong generalization performance. Our code is available at https://anonymous.4open.science/r/MH-pFL.

## 1 INTRODUCTION

Federated learning (FL) has been widely applied in various fields, such as intelligent transportation (Wang et al., 2022; Zhou et al., 2024), healthcare (Nguyen et al., 2022; Murmu et al., 2024), and recommendation systems (Guo et al., 2021; Yuan et al., 2023; Feng et al., 2024). However, a single global model cannot meet all clients' needs due to non-IID data. To address this, personalized federated learning (pFL) (Smith et al., 2017; T Dinh et al., 2020; Deng et al., 2020) emerges, aiming to craft personalized models for clients while enabling knowledge sharing under cross-device settings (McMahan et al., 2016), thus better matching their specific tasks and data distributions.

In practice, devices participating in pFL are often heterogeneous, as they usually have different computational resources (Chai et al., 2020; Shin et al., 2024b), communication capabilities (Caldas et al., 2018; He et al., 2020; Shah & Lau, 2021), and model architectures (Li & Wang, 2019; Zhu et al., 2021; Wu et al., 2024), which complicates the challenges that pFL faces in scenarios of model heterogeneity (Chen et al., 2023). To address the limitations of the model heterogeneous pFL (MH-pFL), several methods have been proposed by researchers, including partial training (Diao et al., 2021; Alam et al., 2022; Hong et al., 2022; Shen et al., 2024), federated distillation (Zhang et al., 2023; Wang et al., 2024; Liu et al., 2025; Guo et al., 2025), and model decoupling (Xu et al., 2023; Collins et al., 2021; Arivazhagan et al., 2019; Liang et al., 2020).

However, modeling decoupling (Jang et al., 2023; Yi et al., 2023a;b) separates local models into feature extractors and classifiers; this low-level knowledge sharing may hinder client collaboration and negatively impact performance. Partial training methods (Horváth et al., 2021; Ilhan et al., 2023; Lee et al., 2024) allow clients to select sub-models for local training. However, differences in model architectures, data distributions, and resource conditions can lead to misalignment in parameter counts

and feature spaces, causing suboptimal performance. Current federated distillation approaches (Gong et al., 2021; Itahara et al., 2021; Wang et al., 2025) depend on a universal or synthetic dataset for knowledge integration. Such additional datasets may limit the applicability of the method and make the performance dependent on the quality of these datasets. Hence, an MH-pFL approach is needed to resolve these issues while ensuring privacy and computational efficiency effectively.

In this work, we revisit hypernetwork (Ha et al., 2017), a model that generates personalized parameters for another neural network, and propose a novel MH-pFL framework, the ***Model-Heterogeneous Personalized Federated HyperNetwork*** (`MH-pFedHN`), which uses a hypernetwork as a knowledge aggregator to enable knowledge fusion among heterogeneous clients. In our approach, MH-pFedHN customizes embedding vectors for each client based on the number of parameters required by the client's model, whereas clients with a similar number of parameters use the same customized embedding vectors. In addition to the feature extractor from traditional hypernetwork shared across all clients to support generalizable feature learning, a shared head is created for clients with the same embedding vectors to generate client-specific parameters for heterogeneous models, which allows the server to generate parameters for multiple models with a similar number of parameters in a single pass, increasing efficiency and promoting knowledge fusion. This fine-grained mapping mechanism enhances the server's expressive and generalization abilities.

To further learn cross-client knowledge and enhance performance, we propose ***MH-pFedHN** with **Global Distillation*** (`MH-pFedHNGD`), with an additional lightweight but effective global model. This plug-in component on the server side is directly generated by our hypernetwork using the global customized embedding vectors. Compared to MH-pFedHN, MH-pFedHNGD leverages a global model to aggregate updates from clients, introducing one more round of lightweight update mechanisms, enabling the hypernetwork to learn a more comprehensive data distribution and more generalizable features. Meanwhile, the global model can serve as a teacher model (Hinton, 2015; Jeong et al., 2018) to assist training via knowledge distillation on the client side, thus improving knowledge acquisition abilities of client-specific models and balancing personalization with generalization.

Both of our methods are data-free and preserve the structural privacy of personalized models. They are the first efficient frameworks designed to leverage hypernetworks for solving MH-pFL problems, and hold great promise for future applications. Our main contributions are as follows.

- We propose MH-pFedHN, a personalized FL method based on hypernetworks and specifically designed for heterogeneous models. This method allows the server to utilize customized embedding vectors and the shared head tailored to clients' needs to generate parameters without disclosing the model architecture to the server, thereby enhancing privacy protection.
- We propose MH-pFedHNGD, which integrates a plug-in component lightweight global model. This approach enhances the hypernetwork's learning and generalization capabilities and allows personalized client models to learn from a global model with knowledge distillation efficiently, resulting in significant performance improvements.
- We evaluate our MH-pFedHN and MH-pFedHNGD for multiple tasks on four popular datasets. The experiments demonstrate that our method exceeds state-of-the-art performance against baselines across all tasks, highlighting the effectiveness and generalizability.

## 2 RELATED WORK

### 2.1 PERSONALIZED FEDERATED LEARNING WITH HYPERNETWORKS

Hypernetworks are methods that use one network to generate weights for other neural networks (Ha et al., 2017), which are widely used in various applications (Suarez, 2017; Nirkin et al., 2021; Beck et al., 2024; Ruiz et al., 2024). In pFL, hypernetworks are used to generate personalized model parameters from clients' embedding vectors (Shamsian et al., 2021; Zhu et al., 2023; Scott et al., 2024) and output the weight ratio during aggregation (Ma et al., 2022). This method has shown effectiveness in systems with diverse data and few-shot learning scenarios (Sendera et al., 2023).

However, these methods are mainly designed to alleviate the problem of statistical heterogeneity, so that clients can obtain personalized models. The work (Shamsian et al., 2021) proposes pFedHN and only explores generating limited heterogeneous models. FLHA-GHN (Litany et al., 2022) uses graph hypernetworks to generate model parameters for different architectures. Training the hypernetwork

on local clients imposes additional computational overhead. They cannot be deployed in resource-constrained environments; in contrast, our efficient methods, MH-pFedHN and MH-pFedHNGD, demonstrate strong potential for deployment in practical scenarios.

## 2.2 MODEL-HETEROGENEOUS PERSONALIZED FEDERATED LEARNING METHODS

Due to resource constraints (Chai et al., 2020; Shin et al., 2024b), communication limitations (Caldas et al., 2018; He et al., 2020; Shah & Lau, 2021), communication limitations, and personalized requirements for models (Li & Wang, 2019; Zhu et al., 2021; Wu et al., 2024), MH-pFL has become an important research direction. Even when clients share the same model architecture, two forms of model heterogeneity can still arise: 1) Due to resource differences, clients may need to employ different sub-models for training (Diao et al., 2021); 2) Clients with varying personalization requirements retain parameters of specific layers solely for local updates (Li et al., 2021), resulting in model heterogeneity. Furthermore, the model architecture may impact model privacy (Zhang et al., 2024a), clients are often reluctant to further disclose the details of their model design information.

**Partial Training** is adopted in some methods, where each client selects a sub-model that originates with the global model. The parameters from the client sub-models are aggregated by averaging the corresponding parameters. HeteroFL (Diao et al., 2021) divides the global model into different sub-models, allowing clients to train appropriate models based on computational capabilities, enabling low-resource clients to contribute to the global model without being excluded from the aggregation. DepthFL (Kim et al., 2023) utilizes different depths of different local clients and prunes the deepest layers off the global model to help allocate the server model to all the clients based on computational resources. pFedGate (Chen et al., 2023) explores to learn a sparse local model adaptively. HypeMeFed (Shin et al., 2024b) combines a multi-path network architecture with weight generation to support client heterogeneity. However, due to differences in model architecture, parameters of different sub-models learning by partial training may not align, resulting in suboptimal performance.

**Federated Distillation** utilizes knowledge distillation (Hinton, 2015) to reduce the impact of data heterogeneity on performance and address model heterogeneity when facilitating collaboration between clients with different architectures by transferring knowledge from complex models to simpler ones in FL (Jeong et al., 2018; Sattler et al., 2020; Wu et al., 2021).

FedMD (Li & Wang, 2019) and DS-FL (Itahara et al., 2021) use a pre-existing public dataset for knowledge extraction, aggregating local soft predictions on the server. KT-pFL (Zhang et al., 2021) trains personalized soft prediction weights on the server to improve heterogeneous models' performance further. Inspired by data-free knowledge distillation (Zhang et al., 2022; Dai et al., 2024), some methods (Zhu et al., 2021; Zhang et al., 2023) use GAN (Goodfellow et al., 2020) to generate synthetic data on the server side. DFRD (Luo et al., 2024) and FedGD (Zhang et al., 2023) design a distributed GAN between the server and the clients to enable data-free knowledge transfer and effectively tackle model heterogeneity. However, training the key generator in GAN-based federated distillation poses significant challenges that can impact the overall performance of the system. FedAKT (Liu et al., 2025) combines knowledge distillation and model decoupling ; however, homogeneous adapters increase computational and communication overhead for clients, and distillation and dual-head mechanisms can perform poorly in highly heterogeneous environments. Therefore, federated distillation still faces the challenges related to privacy risks and the quality of public and synthetic datasets.

**Model Decoupling** is another attempt, which divides client models into feature extractors and classifier heads. Further, scientists extend the previous model decoupling methods to share part of the global model while retaining the other part in a heterogeneous form locally. FedClassAvg (Jang et al., 2023) aggregates classifier weights to establish a consensus on decision boundaries in the feature space, enabling clients with non-IID data to learn from scarce labels; FedGH (Yi et al., 2023a) trains a shared, generalized global prediction head using representations extracted by clients' heterogeneous feature extractors on the FL server. In pFedES (Yi et al., 2023b), clients are trained via their proposed iterative learning method to facilitate the exchange of global knowledge; small local homogeneous extractors are then uploaded for aggregation to help the server learn the knowledge across clients. However, these basic mechanisms might find it difficult to achieve effective knowledge fusion in complex heterogeneous scenarios, which results in limitations in final performance.

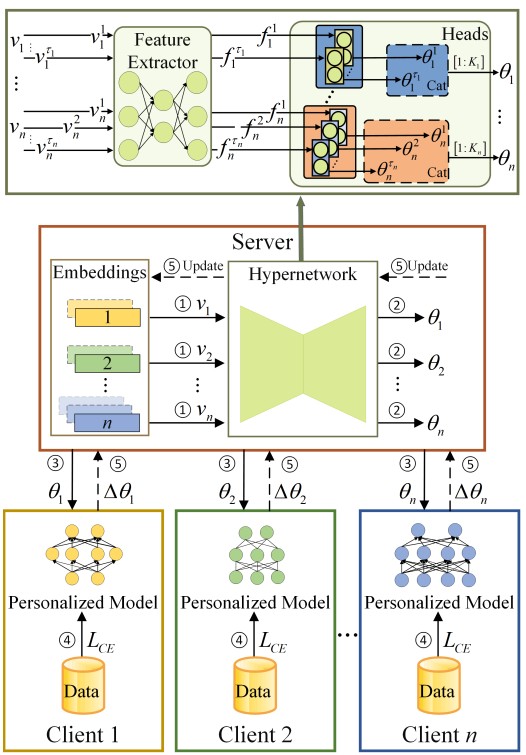

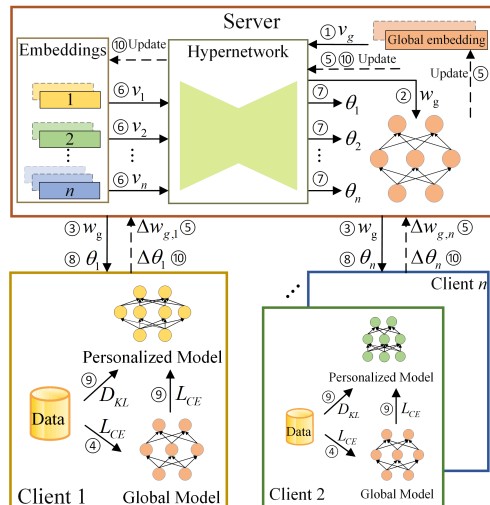

Figure 1: Framework of MH-pFedHN. The workflow contains 5 steps: ① input the embedding matrix $v_i$ of client $i$ into the hypernetwork; ② generates the parameters $\theta_i$ of client $i$; ③ client $i$ receives the parameters $\theta_i$ from the server; ④ client $i$ trains the personalized model $\theta_i$ using the private data; ⑤ client $i$ uploads the update of parameters $\Delta\theta_i$ to the server; server updates customized embedding vectors and hypernetworks.

Figure 2: Framework of MH-pFedHNGD. The workflow contains 10 steps: ① input the embedding matrix $v_g$ of the global model into the hypernetwork; ② generates parameters $w_g$ of the global model; ③ each client receives the global model parameters $w_g$ from the server; ④ each client trains the global model using private data; ⑤ The server receives $\Delta w_{g,i}$ from each client and updates the hypernetwork and global embedding vector parameters; ⑥ input the embedding matrix $v_i$ of client $i$ into the hypernetwork; ⑦ generates parameters $\theta_i$ of client $i$; ⑧ client $i$ receives parameters $\theta_i$ from the server; ⑨ global model $w_g$ assists in training personalized model $\theta_i$ of client $i$ on private data; ⑩ client $i$ uploads update of parameters $\Delta\theta_i$ to the server; server updates customized embedding vectors and hypernetworks.

## 3 OUR FRAMEWORK

In this section, we first formalize the MH-pFL problem. Then, we introduce MH-pFedHN, which generates parameters for different models through customized embedding vectors and utilizes a hypernetwork for knowledge fusion across heterogeneous clients. Finally, we present MH-pFedHNGD, which utilizes a lightweight global model to enhance the learning and generalization capabilities of the hypernetwork. The lightweight plug-in component also assists personalized model training for clients via knowledge distillation, further improving model training across heterogeneous clients. The overviews of our two methods are shown in Figures 1 and 2, where one round of MH-pFedHN consists of 5 steps and one round of MH-pFedHNGD is made up of 10 steps.

### 3.1 PROBLEM FORMULATION

Our goal is to develop an MH-pFL approach that, without requiring knowledge of model architectures, meets the specific requirements of each client to address the challenges brought about by the heterogeneity of the model and data. This can be turned into the following minimization problem, which aims to capture the personalized objectives of each client while accommodating heterogeneous data distributions and model architectures

$$\{\boldsymbol{\theta}_1^*, \ldots, \boldsymbol{\theta}_n^*\} = \underset{\boldsymbol{\theta}_1, \ldots, \boldsymbol{\theta}_n}{\arg\min} \sum_{i=1}^{n} \mathbb{E}_{x,y \sim P_i} [\ell_i(x, y; \boldsymbol{\theta}_i)], \tag{1}$$

where $P_i$ represents the local data distribution of the $i$-th client, and $\ell_i(x_j, y_j; \boldsymbol{\theta}_i)$ denotes the loss function for the $i$-th client's model with parameters $\boldsymbol{\theta}_i$. In particular, $\boldsymbol{\theta}_i$ is specific to each client and can correspond to models of varying architectures.

To further formalize the optimization problem, the training objective is expressed as

$$\arg\min_{\boldsymbol{\theta}_1,\ldots,\boldsymbol{\theta}_n} \sum_{i=1}^{n} L_i(\boldsymbol{\theta}_i) = \arg\min_{\boldsymbol{\theta}_1,\ldots,\boldsymbol{\theta}_n} \sum_{i=1}^{n} \frac{1}{m_i} \sum_{j=1}^{m_i} \ell_i(x_j, y_j; \boldsymbol{\theta}_i), \tag{2}$$

where $m_i$ denotes the number of data samples in the local dataset of the $i$-th client, and $L_i(\boldsymbol{\theta}_i)$ represents the empirical loss over its dataset. By allowing each client to optimize its model, ensuring that the learned parameters $\boldsymbol{\theta}_i$ are suited to the client's unique data distribution and task.

## 3.2 MH-PFEDHN

Here, we describe MH-pFedHN to the MH-pFL problem (Equation 2), and the complete algorithm for MH-pFedHN is provided in Algorithm 1. Let $h(\cdot; \boldsymbol{\varphi})$ denote the hypernetwork $h$ with parameters $\boldsymbol{\varphi}$, where $\boldsymbol{\varphi}$ is composed by a feature extractor $\boldsymbol{\varphi}_f$ and multiple heads $\{\boldsymbol{\varphi}_{H_l}\}$.

In the initial stage, clients need to upload the number of parameters $K$ required for their personalized heterogeneous models. The server then dynamically determines the hypernetwork output dimension $N$ based on $K$ (we recommend that the value of $\lceil K_i/N \rceil$ should be greater than the number of layers in the client model) and calculates the customized number of embedding vectors needed for $i$-th client as $\tau_i = \lceil K_i/N \rceil$. Therefore, models with similar parameters share the same number of customized embedding vectors. These personalized heterogeneous models share the same established head $\boldsymbol{\varphi}_{H_l}$.

*Specifically*, for example, if two models have the same number of parameters and are both determined to have three customized embedding vectors, they will share the same head $\boldsymbol{\varphi}_{H_l}$. This head has three output channels (the channel number equals the number of customized embedding vectors), each tailored for a specific customized embedding vector input. The shared feature extractor $\boldsymbol{\varphi}_f$ for all clients and this head $\boldsymbol{\varphi}_{H_l}$ will generate three subsets of parameters with length $N$ for both models. These three subsets of parameters, when combined and rounded down according to the specific parameter size of each client, constitute the personalized parameters for each client. *Generally*, for the $j$-th customized embedding vector of client $i$ (associated with the $l$-th head), the output from the feature extractor is processed as $\boldsymbol{\theta}_i^j = h(\boldsymbol{v}_i^j; \boldsymbol{\varphi}_f, \boldsymbol{\varphi}_{H_l})$, where we use $\boldsymbol{v}_i = [\boldsymbol{v}_i^1, \ldots, \boldsymbol{v}_i^{\tau_i}]$ to denotes the customized embedding vectors for the $i$-th client $\boldsymbol{\theta}_i = \boldsymbol{\theta}_i(\boldsymbol{\varphi}) := h(\boldsymbol{v}_i; \boldsymbol{\varphi})_{[1:K_i]}$. *Finally*, the personalized model parameters for client $i$ are generated as follows:

$$\boldsymbol{\theta}_i := \text{concat}(\boldsymbol{\theta}_i^1, \boldsymbol{\theta}_i^2, \cdots, \boldsymbol{\theta}_i^{\tau_i})_{[1:K_i]}, \tag{3}$$

After client $i$ trains the personalized model based on their private data, the hypernetwork $h$ updates a set of $\Delta\boldsymbol{\theta}_i$. Therefore, our MH-pFedHN allows the server to generate parameters for multiple models with a similar number of parameters in a single pass.

Based on the aforementioned setup, we adopt the MH-pFL objective (Equation 2) and get our MH-pFedHN optimization function as[1]:

$$\arg\min_{\boldsymbol{\varphi}, \boldsymbol{v}_1, \ldots, \boldsymbol{v}_n} \sum_{i=1}^{n} L_i\left(h(\boldsymbol{v}_i; \boldsymbol{\varphi})_{[1:K_i]}\right) = \arg\min_{\boldsymbol{\varphi}, \boldsymbol{v}_1, \ldots, \boldsymbol{v}_n} \sum_{i=1}^{n} \frac{1}{m_i} \sum_{j=1}^{m_i} \left(\ell_i(x_j, y_j; \left(h(\boldsymbol{v}_i; \boldsymbol{\varphi})_{[1:K_i]}\right)\right)$$

$$= \arg\min_{\boldsymbol{\theta}_1, \ldots, \boldsymbol{\theta}_n} \sum_{i=1}^{n} \frac{1}{m_i} \sum_{j=1}^{m_i} \ell_i(x_j, y_j; \boldsymbol{\theta}_i), \tag{4}$$

## 3.3 MH-PFEDHNGD

To enhance the hypernetwork's learning capacity and generalization ability while enabling clients to efficiently extract knowledge representations from a global model, we propose MH-pFedHNGD, which introduces a lightweight global model based on MH-pFedHN. This design aims to improve overall performance at the cost of minimal system and communication overhead. The complete algorithm for MH-pFedHNGD can be found in Algorithm 2.

---

[1]The theoretical analysis is in Appendix B.

Thus, the global model's number of parameters is set as $K_g = \min\{K_1, \cdots, K_n\}$, thus, the global model shares the same head with the client having the fewest number of parameters to make sure the global model lightweight. Then the number of global customized embedding vectors is set as $\tau_g = \lceil K_g/N \rceil$, the global embedding vector $\boldsymbol{v}_g = [\boldsymbol{v}_g^1, \ldots, \boldsymbol{v}_g^{\tau_g}]$. Assuming the client with the fewest number of parameters corresponds to the $l$-th head, the global model parameters are generated by:

$$\boldsymbol{w}_g := h(\boldsymbol{v}_g; \boldsymbol{\varphi})_{[1:K_g]} = \text{concat}\left(h(\boldsymbol{v}_g^1; \boldsymbol{\varphi}_f, \boldsymbol{\varphi}_{H_{k_l}}), h(\boldsymbol{v}_g^2; \boldsymbol{\varphi}_f, \boldsymbol{\varphi}_{H_{k_l}}), \cdots, h(\boldsymbol{v}_g^{\tau_g}; \boldsymbol{\varphi}_f, \boldsymbol{\varphi}_{H_{k_l}})\right)_{[1:K_g]}, \quad (5)$$

Then, each client receives the parameters $\boldsymbol{w}_{g,i} = \boldsymbol{w}_g$ and trains on private data, which is to optimize:

$$\arg\min_{\boldsymbol{\varphi}, \boldsymbol{v}_g} \sum_{i=1}^n \frac{m_i}{M} L_i\left(h(\boldsymbol{v}_g; \boldsymbol{\varphi})_{[1:K_g]}\right) = \arg\min_{w_i} \sum_{i=1}^n \frac{m_i}{M} L_i(\boldsymbol{w}_{g,i}), \quad (6)$$

$$L_i(\boldsymbol{w}_{g,i}) = \frac{1}{m_i} \sum_{j=1}^{m_i} \ell_i(x_j, y_j; \boldsymbol{w}_{g,i}), \quad (7)$$

where $M = \sum_i m_i$. After training its own global model completed in Step④ in Figure 2, clients upload $\Delta \boldsymbol{w}_{g,i} = \boldsymbol{w}_{g,i} - \boldsymbol{w}_g$ to server to update the hypernetwork one more time along with global customized embedding vectors, which enhances the learning and generalization capabilities of the hypernetwork.

Finally, in the distillation training phase, client $i$ reloads the global model in Step③ and receives the parameters $\boldsymbol{\theta}_i$ of the personalized model, utilizing the global model as a teacher model, directing the training of the personalized model for client $i$. We adopt the optimization problem (Equation 4) and get our MH-pFedHNG optimization function as follows:

$$\arg\min_{\boldsymbol{\varphi}, \boldsymbol{v}_1, \ldots, \boldsymbol{v}_n} \sum_{i=1}^n \left[\lambda L_i\left(h(\boldsymbol{v}_i; \boldsymbol{\varphi})_{[1:K_i]}\right) + (1-\lambda) L_{KL}\left(h(\boldsymbol{v}_i; \boldsymbol{\varphi})_{[1:K_i]}, h(\boldsymbol{v}_g; \boldsymbol{\varphi})_{[1:K_g]}\right)\right], \quad (8)$$

where $L_{KL}$ is the Kullback–Leibler divergence (Csiszár, 1975), $\lambda$ is a hyperparameter used to balance the distillation loss and the cross-entropy loss. Therefore, the lightweight plug-in component further improves model training across heterogeneous clients.

## 4 EXPERIMENTS

### 4.1 EXPERIMENT SETUP

**Datasets.** We evaluate the MH-pFedHN and MH-pFedHNGD framework over four datasets, EMNIST, CIFAR-10, CIFAR-100, and Tiny-ImageNet. We adopt two non-IID settings (T Dinh et al., 2020; Liu et al., 2024).1) Quantity-based label imbalance (non-IID_1). For the EMNIST, CIFAR-100, and Tiny-ImageNet datasets, we randomly allocate 6, 10, and 20 classes to each client, respectively. We draw $\alpha_{i,c} \sim U(0.4, 0.6)$, and allocate $\frac{\alpha_{i,c}}{\sum_j \alpha_{j,c}}$ of the samples for the class $c$ selected on client $i$. 2) Distribution-based label imbalance (non-IID_2). We employ the Dirichlet distribution $Dir(0.01)$ to partition the dataset among the clients. For each client, 75% of the data is used for training, and 25% is used for testing. As EMNIST dataset is simpler than others, we only report results for 200 clients.

**Models.** For a fair comparison, we use the LeNet-style model (LeCun et al., 1998) in the homogeneous model experiments as prior students. In addition, we use a VGGNet (Simonyan & Zisserman, 2015), along with three residual networks (He et al., 2016).[2] All of our heterogeneous experiments use these five models, which are evenly distributed among all clients by default. The feature extractor of the hypernetwork is comprised by a three-layer fully connected network. The details are in Appendix G.

**Baselines.** We choose various state-of-the-art. **pFedHN** (Shamsian et al., 2021) uses hypernetwork to produce personalized model directly; **pFedLA** (Ma et al., 2022) computes aggregation weights for local models of each client; **FedGH** (Yi et al., 2023a) uses a generalized global prediction header for diverse model structures; **pFedLHN** (Zhu et al., 2023) leverages a layer-wise hypernetwork

---

[2]Experiments with Vision Transformer (ViT) are in Appendix D.2, which shows that our method supports large models. Experiments with CIFAR-10 are in Appendix D.4.

Table 1: Homogeneous model experiments, where left side is non-IID_1 and right is non-IID_2.

| | CIFAR-100 | | | | | | Tiny-ImageNet | | | | | | EMNIST | |
|---|---|---|---|---|---|---|---|---|---|---|---|---|---|---|
| # Clients | 50 | | 100 | | 200 | | 50 | | 100 | | 200 | | 200 | |
| Local | 50.32 | 72.79 | 40.41 | 73.52 | 34.65 | 73.56 | 29.17 | 50.65 | 20.73 | 55.14 | 14.90 | 55.81 | 96.26 | 98.70 |
| FedAvg (McMahan et al., 2017) | 22.94 | 22.59 | 24.80 | 24.27 | 25.55 | 23.07 | 8.13 | 6.64 | 8.80 | 7.54 | 9.12 | 8.22 | 81.49 | 80.18 |
| pFedHN (Shamsian et al., 2021) | 63.66 | 76.76 | 58.90 | 79.74 | 32.77 | 74.14 | 40.10 | 56.76 | 35.39 | 58.93 | 32.10 | 61.12 | 97.50 | 99.18 |
| pFedLA (Ma et al., 2022) | 63.33 | 72.86 | 55.83 | 75.22 | 55.36 | 75.88 | 39.69 | 48.30 | 30.00 | 53.59 | 23.42 | 54.84 | 95.41 | 98.40 |
| FedGH (Yi et al., 2023a) | 61.01 | 75.54 | 53.61 | 76.65 | 38.70 | 77.30 | 31.67 | 54.30 | 23.64 | 57.97 | 17.80 | 57.44 | 96.17 | 96.42 |
| pFedLHN (Zhu et al., 2023) | 61.00 | 77.03 | 58.39 | 79.06 | 53.43 | 79.49 | 39.49 | 55.71 | 35.00 | 60.62 | 31.31 | 58.30 | 97.52 | **99.25** |
| PeFLL (Scott et al., 2024) | 50.64 | 66.79 | 49.20 | 71.95 | 40.97 | 74.11 | 32.41 | 51.44 | 28.07 | 55.86 | 23.65 | 56.26 | 94.68 | 98.88 |
| FedAKT (Liu et al., 2025) | 60.85 | 74.28 | 56.09 | 77.92 | 51.48 | 78.56 | 38.57 | 55.23 | 31.86 | 60.23 | 31.87 | 61.58 | 97.21 | 98.96 |
| MH-pFedHN (ours) | **64.69** | 77.93 | **63.32** | 80.93 | 60.11 | 81.60 | 42.35 | 58.30 | 37.62 | 62.68 | 37.30 | 63.61 | 97.56 | 99.16 |
| MH-pFedHNGD (ours) | **68.30** | **80.07** | **63.97** | **82.51** | **61.59** | **82.54** | **44.44** | **58.35** | **40.99** | **63.39** | **39.96** | **66.67** | **97.76** | 98.83 |

Table 2: Heterogeneous model experiments, where left side is non-IID_1 and right is non-IID_2.

| | CIFAR-100 | | | | | | Tiny-ImageNet | | | | | | EMNIST | |
|---|---|---|---|---|---|---|---|---|---|---|---|---|---|---|
| # Clients | 50 | | 100 | | 200 | | 50 | | 100 | | 200 | | 200 | |
| Local | 50.74 | 71.69 | 40.41 | 72.01 | 31.95 | 73.05 | 29.64 | 51.64 | 19.63 | 54.05 | 14.44 | 53.92 | 96.77 | 98.83 |
| FedAvg (McMahan et al., 2017) | 22.80 | 17.27 | 24.72 | 22.18 | 24.28 | 21.29 | 8.17 | 8.22 | 8.59 | 11.86 | 9.11 | 16.40 | 85.09 | 84.78 |
| pFedHN (Shamsian et al., 2021) | 50.93 | 73.22 | 42.82 | 74.73 | 12.49 | 72.62 | 32.15 | 54.29 | 24.33 | 58.31 | 15.64 | 57.82 | 97.39 | 99.07 |
| pFedLA (Ma et al., 2022) | 51.97 | 71.24 | 52.06 | 74.68 | 40.11 | 75.24 | 26.97 | 46.86 | 19.70 | 51.73 | 17.28 | 54.44 | 93.82 | 98.36 |
| FedGH (Yi et al., 2023a) | 52.61 | 71.38 | 42.45 | 73.25 | 31.57 | 71.57 | 25.69 | 50.73 | 16.90 | 53.57 | 11.74 | 55.04 | 96.01 | 96.37 |
| pFedLHN (Zhu et al., 2023) | 55.91 | 75.28 | 48.46 | 76.16 | 40.96 | 74.60 | 38.13 | 55.13 | 29.20 | 59.36 | 20.85 | 57.53 | 97.47 | 99.15 |
| PeFLL (Scott et al., 2024) | 55.87 | 72.28 | 52.09 | 74.51 | 42.56 | 72.79 | 35.70 | 55.21 | 30.08 | 55.60 | 25.24 | 55.01 | 97.46 | 99.10 |
| FedAKT (Liu et al., 2025) | 53.28 | 73.15 | 43.80 | 75.25 | 35.37 | 74.98 | 35.57 | 53.74 | 28.25 | 56.95 | 20.17 | 55.36 | 97.43 | 99.11 |
| MH-pFedHN (ours) | **57.09** | **75.40** | 50.35 | **77.24** | **42.98** | 75.38 | 38.00 | **55.51** | **30.32** | 58.93 | 23.31 | **58.79** | 97.58 | 99.18 |
| MH-pFedHNGD (ours) | **60.11** | **76.76** | **52.14** | 77.03 | **43.41** | 76.46 | **42.18** | **58.02** | **34.98** | **61.11** | **26.12** | **60.38** | 97.65 | 98.73 |

to achieve fine-grained personalization across different layers; **PeFLL** (Scott et al., 2024) uses a learning-to-learn approach to generate models; **FedAKT** (Liu et al., 2025) employs adapters and distillation to enhance knowledge transfer and model adaptation in heterogeneous environments. We also include **Local** Training without aggregation and **FedAvg** (McMahan et al., 2017).

**Training Strategies.** We have at most 500 server-client communication rounds, with results averaged over three runs, as the standard deviation is usually smaller than 0.2%, and we omit the error bar. For MH-pFedHN, we set local epochs to 2, SGD optimizer with the learning rate $1e-3$ and weight decay $1e-4$ and momentum 0.9, batch size 64. The hypernetwork uses the Adam optimizer (Kingma & Ba, 2014) with a learning rate of $2e-4$, embedding dimension 64, and the output dimension 3072. For MH-pFedHNGD, we configure the global model LeNet-5. On CIFAR-100, Tiny-ImageNet, and EMNIST datasets, we set distillation temperatures as 15, 24, and 10, and distillation loss coefficients as 0.01, 0.2, and 0.1, respectively. More settings and design choices are in Appendix C and F.

## 4.2 HOMOGENEOUS MODEL EXPERIMENTS

The results in Table 1 demonstrate that our proposed methods, MH-pFedHN and MH-pFedHNGD, both outperform all the baseline approaches under both non-IID scenarios. Although the accuracy of all methods decreases as the number of clients increases, our methods consistently maintain high accuracy, underscoring their robustness in handling data heterogeneity and scalability to many clients. MH-pFedHNGD achieves consistently higher test accuracy compared to MH-pFedHN, further validating the effectiveness of generating a global model for the hypernetwork update and leveraging it for distillation training. Furthermore, most methods show high test accuracy (up to 96%) on the EMNIST dataset due to its simplicity. However, this feature limits observable performance differences, making meaningful comparisons challenging on the EMNIST dataset. In contrast, our method is better equipped to handle complex real-world data scenarios.

## 4.3 HETEROGENEOUS MODEL EXPERIMENTS

For pFedHN and pFedLHN, we modify them by adding multiple heads, enabling them to generate parameters for different models for heterogeneous model experiments. As PeFLL, pFedLA, FedAvg, and Local Training do not allow model heterogeneity, we thus conduct repeat experiments using various models and take the average test accuracy as the final results.

Experiment results in Table 2 indicate that methods that allow model heterogeneity show a decline in accuracy under the same settings in most cases for homogeneous models. This suggests that collaboration among heterogeneous clients is still insufficient and challenging, an issue we need to address in the future. For PeFLL, pFedLA, FedAvg, and Local Training that do not allow model heterogeneity, the results are average based on LeNet, VGGNet, and ResNet, which may be better than

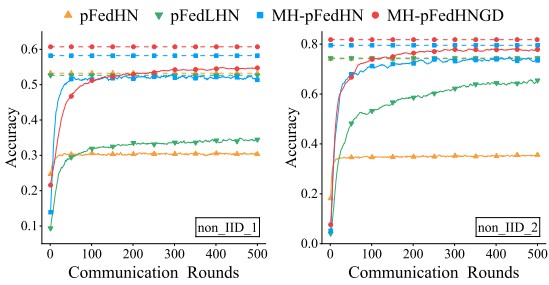 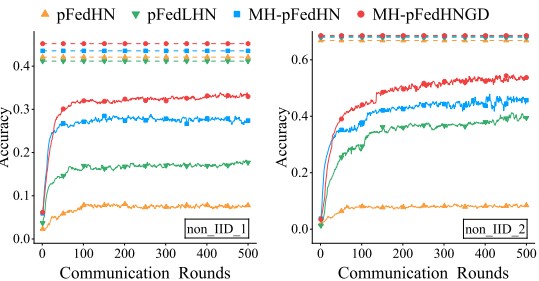

Figure 3: Generalization experiments (homogeneous), where solid lines denote test clients and dashed lines denote training clients.

Figure 4: Generalization experiments (heterogeneous), where solid lines denote test clients and dashed lines denote training clients.

the results in Table 1 as they only have LeNet. Overall, our two approaches still outperform all the baseline methods. Furthermore, MH-pFedHNGD shows significant improvement over MH-pFedHN in most settings, highlighting the necessity of introducing a global model and conducting knowledge distillation training in settings with model heterogeneity.

## 4.4 EXPERIMENT WITH GENERALIZATION

Generalization in FL refers to the model's ability to perform well on unseen clients. This section evaluates the generalization capabilities of MH-pFedHN and MH-pFedHNGD. These experiments utilize the CIFAR-100 dataset with 100 clients, of which 80 clients are used to train the hypernetwork, while the remaining 20 clients are designated for generalization testing. During the testing phase, the parameters of the hypernetwork remain fixed, and only the customized embedding vectors corresponding to each client are optimized.

**Generalize to homogeneous clients.** Figure 3 shows that MH-pFedHNGD and MH-pFedHN exhibit outstanding generalization capabilities, significantly outperforming other baseline methods up to 20%. In the early stages of generalization, both algorithms converge rapidly and achieve accuracy on the test set comparable to that of the training clients. Moreover, the generalization ability of MH-pFedHNGD is superior to that of MH-pFedHN, indicating that introducing a global model is beneficial for further enhancing the generalization performance of the hypernetwork.

**Generalize to heterogeneous clients.** In the experiments with heterogeneous models, we used the LeNet, VGGNet, and ResNet. We then added the MLP model and SqueezeNet1_0 model (Iandola, 2016). These five models were evenly distributed among clients. Figure 4 shows that our method still exhibits the better generalization performance (up to 50%) of all the baselines for heterogeneous models, significantly outperforming the other two baselines. Again, the generalization ability of MH-pFedHNGD is notably superior to MH-pFedHN.

**Generalize to new architectures.** All the baselines can not handle this challenge. We use new ResNet and SqueezeNet1_1 for the added clients, while the training models are all different architectures. In this case, only the feature extractor of the hypernetwork is frozen; the new head will be added and trained. Figure 5 shows no significant difference between MH-pFedHN and MH-pFedHNGD. The assistance provided by the global model is limited, and the learning during the generalization process primarily relies on the data distribution itself. Still, our methods outperform baselines.

## 4.5 ABLATION STUDY

First, we investigate the head components of MH-pFedHN. We have two configurations: (1) MH-pFedHN does not have any heads; the hypernetwork consists of only a feature extractor; (2) MH-pFedHN has one head with an output dimension of $N$. Table 3 indicates that without the head, the performance degrades significantly, demonstrating the importance of our head architecture in generating effective parameters. When using only a single head, most settings encounter out-of-memory issues, and the rest two cases show limited performance, which highlights that our carefully designed shared head can greatly reduce computation overhead and hold strong practical potential.

Next, we examine the global model. We use MH-pFedHNG to denote MH-pFedHNGD without knowledge distillation. Table 4 shows that in most cases, MH-pFedHNGD outperforms MH-pFedHNG,

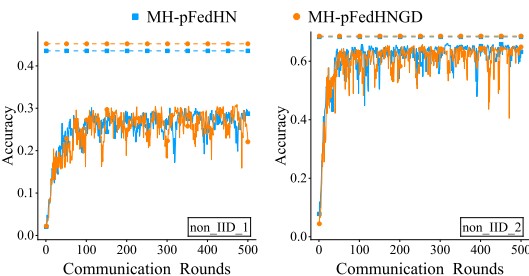

Figure 5: Generalization experiments for new architectures, where solid lines denote test clients and dashed lines denote training clients.

Table 3: The ablation experiments of MH-pFedHN. Green indicates the setting without a head, and Red indicates the setting where each client has one head. "-" indicates out of memory.

| Data_Distribution | CIFAR-100 | | | | | |
| | non-IID_1 | | | non-IID_2 | | |
| # Clients | 50 | 100 | 200 | 50 | 100 | 200 |
| homogeneous | 58.32 | 47.86 | 40.63 | 76.01 | 77.26 | 76.76 |
| heterogeneous | 39.72 | 35.40 | 29.28 | 57.74 | 64.27 | 60.68 |
| homogeneous | 53.51 | - | - | 73.70 | - | - |
| heterogeneous | - | - | - | - | - | - |

Table 4: The ablation experiments of MH-pFedHNGD. Green and red correspond to homogeneous and heterogeneous models, respectively.

| Data_Distribution | CIFAR-100 | | | | | |
| | non-IID_1 | | | non-IID_2 | | |
| # Clients | 50 | 100 | 200 | 50 | 100 | 200 |
| MH-pFedHN | 63.91 | 63.14 | 59.25 | 78.25 | 80.79 | 81.04 |
| MH-pFedHNG | 66.19 | 63.36 | 59.71 | 79.18 | 81.80 | 81.85 |
| MH-pFedHNGD | 68.27 | 63.58 | 61.19 | 80.18 | 82.31 | 82.19 |
| MH-pFedHN | 57.18 | 50.35 | 40.83 | 74.84 | 76.78 | 74.71 |
| MH-pFedHNG | 57.01 | 49.81 | 41.45 | 75.44 | 76.32 | 76.04 |
| MH-pFedHNGD | 60.11 | 51.57 | 43.41 | 76.76 | 77.03 | 76.46 |

Table 5: The experiments with different architectures for the global and personalized models. Green indicates MH-pFedHN, red indicates MH-pFedHNGD.

| Data_Distribution | CIFAR-100 | | | | | |
| | non-IID_1 | | | non-IID_2 | | |
| # Clients | 50 | 100 | 200 | 50 | 100 | 200 |
| VGGNet | 60.04 | 50.67 | 39.35 | 77.72 | 77.71 | 76.94 |
| ResNet | 62.03 | 55.38 | 42.09 | 75.57 | 72.50 | 65.50 |
| MLP | 52.14 | 49.09 | 43.48 | 69.97 | 73.13 | 74.32 |
| VGGNet/ResNet/MLP | 56.16 | 48.94 | 42.11 | 73.73 | 75.32 | 75.01 |
| VGGNet | 64.75 | 61.03 | 44.65 | 79.82 | 79.93 | 78.07 |
| ResNet | 64.47 | 55.40 | 49.43 | 75.66 | 74.59 | 73.67 |
| MLP | 52.20 | 49.17 | 45.37 | 69.98 | 73.58 | 74.75 |
| VGGNet/ResNet/MLP | 59.11 | 50.05 | 43.72 | 74.66 | 76.68 | 76.25 |

which in turn outperforms MH-pFedHN. This demonstrates that the global model alone can enhance generalization ability of the hypernetwork, while knowledge distillation with the global model further improves model training across heterogeneous clients. The observation that results in homogeneous settings are better than those in heterogeneous ones is also consistent with other experiments.

## 4.6 EXPERIMENTS WITH DIFFERENT ARCHITECTURES

In the previous experiments with heterogeneous models, some clients and the global model also used the same LeNet-style architecture. Therefore, we further investigate the scenario where the architectures of the global model and the personalized client models are completely different. To this end, we set up experiments where the personalized client models are all VGGNet, MLP, and ResNet, while the global model is the LeNet-style model. We also consider the case where these three types of models are evenly distributed among the clients, with the global model still being the LeNet-style model. Table 5 shows that for MH-pFedHNGD, even when the architectures of the global and client models are completely different, its performance still surpasses that of MH-pFedHN, which is consistent with our previous findings. Under such a challenging scenario, both models achieve promising results, demonstrating the strong robustness and generalization ability of our method.

## 4.7 SUPPLEMENTARY EXPERIMENTS

Experiments with CIFAR-10, ViT, scalability, client participation ratios, different architectures of global model, redundant parameter, shared heads for heterogeneous models with similar parameter sizes, more baselines, overhead, communication efficiency, extremely personalized setting, and resource constraint setting are in Appendix D. Experiments with **privacy concerns** (iDLG Attacks and differential privacy) are in Appendix E. Hyperparameter choice experiments are in Appendix F.

## 5 CONCLUSION

In this work, we propose a novel data-free approach for model-heterogeneous personalized federated learning, termed MH-pFedHN. This method employs a hypernetwork to generate the model parameters for each client, allowing clients to customize their network architectures without exposing them to the server. Furthermore, we present an improved MH-pFedHN with minimal effort, denoted as MH-pFedHNGD, which significantly enhances the learning and generalization abilities of the hypernetwork, improves the learning effectiveness of personalized models for clients, and reduces the risk of overfitting. Overall, the effectiveness of our approaches stands out through extensive experiments over other baselines conducted in various settings.

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

# Part I

# Appendix

## Table of Contents

**Algorithm 1** Model-Heterogeneous Personalized Federated Hypernetwork

---

**Input:** $R$ - number of rounds, $\alpha$ - HN learning rate, $\eta$ - client learning rate, $E$ - client local epoch, $\{K_1, \ldots, K_n\}$ - number of clients parameter, $\{D_1, \ldots, D_n\}$ - datasets

**Output:** trained model parameters $\{\theta_1, \ldots, \theta_n\}$

  **procedure** SERVER EXECUTES
    **for** $r = 1$ **to** $R$ **do**
      **for** each client $i$ **in parallel do**
        $\theta_i = h(v_i; \varphi)_{[1:K_i]}$
        $\Delta\theta_i \leftarrow \text{ClientUpdate}(\theta_i)$
        $\varphi = \varphi - \alpha\nabla_\varphi \theta_i^T \Delta\theta_i$
        $v_i = v_i - \alpha\nabla_{v_i}\varphi^T \nabla_\varphi \theta_i^T \Delta\theta_i$
      **end for**
    **end for**
  **end procedure**
  **function** CLIENTUPDATE$(\theta_i)$
    $\widetilde{\theta}_i = \theta_i$
    **for** $e = 1$ **to** $E$ **do**
      sample batch $B \subset D_i$
      $\widetilde{\theta}_i = \widetilde{\theta}_i - \eta\nabla_{\widetilde{\theta}_i}L(\widetilde{\theta}_i, B)$
    **end for**
    $\Delta\theta_i = \widetilde{\theta}_i - \theta_i$
    **return** $\Delta\theta_i$
  **end function**

---

**Algorithm 2** Model-Heterogeneous Personalized Federated Hypernetwork with Global Distillation

---

**Input:** $R$ - number of rounds, $\alpha$ - HN learning rate, $\eta$ - client learning rate, $E$ - client local epoch, $\{K_1, \ldots, K_n\}$ - number of clients parameter, $\{D_1, \ldots, D_n\}$ - datasets, $\lambda$ - balancing factor of distillation loss

**Output:** trained model parameters $\{\theta_1, \ldots, \theta_n\}$

  **procedure** SERVER EXECUTES
    **for** $r = 1$ **to** $R$ **do**
      $w_g = h(v_g; \varphi)_{[1:K_g]}$
      **for** each client $i$ **in parallel do**
        $\Delta w_{g,i} \leftarrow \text{ClientUpdate1}(w_g)$
      **end for**
      $\varphi = \varphi - \alpha\sum_{i=1}^n \frac{m_i}{M}\nabla_\varphi w_g^T \Delta w_{g,i}$
      $v_g = v_g - \alpha\sum_{i=1}^n \frac{m_i}{M}\nabla_{v_g}\varphi^T \nabla_\varphi w_g^T \Delta w_{g,i}$
      **for** each client $i$ **in parallel do**
        $\theta_i = h(v_i; \varphi)_{[1:K_i]}$
        $\Delta\theta_i \leftarrow \text{ClientUpdate2}(\theta_i)$
        $\varphi = \varphi - \alpha\nabla_\varphi \theta_i^T \Delta\theta_i$
        $v_i = v_i - \alpha\nabla_{v_i}\varphi^T \nabla_\varphi \theta_i^T \Delta\theta_i$
      **end for**
    **end for**
  **end procedure**
  **function** CLIENTUPDATE1$(w_g)$
    $w_{g,i} = w_g$
    **for** $e = 1$ **to** $E$ **do**
      sample batch $B \subset D_i$
      $w_{g,i} = w_{g,i} - \eta\nabla_{w_{g,i}}L(w_{g,i}, B)$
    **end for**
    $\Delta w_{g,i} = w_{g,i} - w_g$
    **return** $\Delta w_{g,i}$
  **end function**
  **function** CLIENTUPDATE2$(\theta_i)$
    $\widetilde{\theta}_i = \theta_i$
    **for** $e = 1$ **to** $E$ **do**
      sample batch $B \subset D_i$
      $\widetilde{\theta}_i = \widetilde{\theta}_i - \eta\nabla_{\widetilde{\theta}_i}\left(\lambda L(\widetilde{\theta}_i, B) + (1-\lambda)L_{KL}(\widetilde{\theta}_i, w_g, B)\right)$
    **end for**
    $\Delta\theta_i = \widetilde{\theta}_i - \theta_i$
    **return** $\Delta\theta_i$
  **end function**

---

## A  ALGORITHMS

In this section, we will outline the pseudo-algorithms related to MH-pFedHN and MH-pFedHNGD (see Algorithms 1 and 2).

# B  THEORETICAL ANALYSIS

Here, we provide a theoretical analysis of MH-pFedHN. Given the difficulty of providing proofs under complex network settings, without loss of generality, we begin with a simplified model to establish the effectiveness of our method in achieving the optimal $\bar{\theta}$.

Specifically, we first utilize a simple linear version of the hypernetwork to give further insight into the solution of MH-pFedHN (Equation 4), where the feature extractor $W_f$ and multiple heads $\{W_{H_l}\}$ in the hypernetwork, as well as the target network $\theta$, are all linear models. For the client $i$ associated with the $l$-th head,

$$\theta_i = W_{H_l} W_f v_i;$$

with $W_{H_l} \in \mathbb{R}^{N \times d}$, $W_f \in \mathbb{R}^{d \times k}$ and $v_i \in \mathbb{R}^{k \times \tau_l}$ is the $i$-th client's embeddings.

Let $V_l$ denote the $\tau_l \times k \times n_l$ tensor whose slice are the clients' embedding vectors $v_i$. where $n_l$ is the number of clients corresponding to the $l$-th head, and $\tau_l$ is the number of embedding vectors for those clients.

We note that even for convex loss functions $L_i(\theta)$, the objective is shown as

$$C(\{W_{H_l}\}, W_f, \{V_l\}) = \sum_l \sum_i L_i(W_{H_l} W_f v_i).$$

This objective formula might not be convex in $(\{W_{H_l}\}, W_f, \{V_l\})$ but block multi-convex. In one setting, however, we get a nice analytical solution.

**Proposition 1.** Let $\{X_i, y_i\}$ be the data for the client $i$ and let the loss for linear regressor $\theta_i$ be $L_i(\theta_i) = \|X_i \theta_i - y_i\|^2$. Furthermore, assume that for all $i$, $X_i^T X_i = I_d$. Define the empirical risk minimization (ERM) solution for client $i$ to be $\bar{\theta}_i = \arg\min_{\theta \in \mathbb{R}^d} \|X_i \theta - y_i\|^2$.

The optimal $\{W_H\}, W_f, \{V\}$ minimizing $\sum_l \sum_i \|X_i (W_{H_l} W_f) v_i - y_i\|^2$ are given by PCA on $(\bar{\theta}_1, \ldots, \bar{\theta}_n)$, where $(W_{H_l} W_f)$ is the top-$k$ principal components and $v_i$ is the coefficients for $\bar{\theta}_i$ in these components.

**Proof for Proposition 1.** Let $\bar{\theta}_i$ denote the optimal solution at client $i$, then

$$\bar{\theta}_i = (X_i^T X_i)^{-1} X_i^T y_i = X_i^T y_i.$$

We have

$$\arg\min_{\theta_i} \|X_i \theta_i - y_i\|^2 = \arg\min_{\theta_i} (X_i \theta_i - y_i)^T (X_i \theta_i - y_i).$$

$$= \arg\min_{\theta_i} \theta_i^T X_i^T X_i \theta_i - 2\theta_i^T X_i^T y_i + y_i^T y_i = \arg\min_{\theta_i} \theta_i^T \theta_i - 2\langle \theta_i, \bar{\theta}_i \rangle + \|\bar{\theta}_i\|^2,$$

$$= \arg\min_{\theta_i} \|\theta_i - \bar{\theta}_i\|^2.$$

For the client $i$ associated with the $l$-th head, denote $\theta_i = (W_{H_l} W_f) v_i$.

Our optimization problem becomes

$$\arg\min_{W_{H_l}, W_f, V_l} \sum_i^{n_l} \|(W_{H_l} W_f) v_i - \bar{\theta}_i\|^2.$$

Without loss of generality, we can optimize $W_{H_l} W_f$ over the set of all matrices with orthonormal columns, i.e.,

$$(W_{H_l} W_f)^T (W_{H_l} W_f) = I.$$

Since for each solution $((W_{H_l} W_f), V_l)$, we can obtain the same loss for $((W_{H_l} W_f) R, R^{-1} V_l)$ where $R$ is invertible, we can select an $R$ that performs Gram-Schmidt orthonormalization on the columns of $(W_{H_l} W_f)$.

In the case of fixed $(W_{H_l} W_f)$, the optimal solution for $v_i$ is given by

$$v_i^* = ((W_{H_l} W_f)^T (W_{H_l} W_f))^{-1} (W_{H_l} W_f)^T \bar{\theta}_i = (W_{H_l} W_f)^T \bar{\theta}_i.$$

Hence, our optimization problem becomes

$$\arg\min_{(\boldsymbol{W}_{H_l}\boldsymbol{W}_f);(\boldsymbol{W}_{H_l}\boldsymbol{W}_f)^T(\boldsymbol{W}_{H_l}\boldsymbol{W}_f)=\boldsymbol{I}}\sum_{i}^{n_l}\|(\boldsymbol{W}_{H_l}\boldsymbol{W}_f)(\boldsymbol{W}_{H_l}\boldsymbol{W}_f)^T\bar{\boldsymbol{\theta}}_i-\bar{\boldsymbol{\theta}}_i\|^2,$$

which is equivalent to performing PCA on $\{\bar{\boldsymbol{\theta}}_i\}_1^{n_l}$.

This demonstrates that our method can obtain the optimal $\bar{\boldsymbol{\theta}}$, thereby validating **Proposition 1**.

## C    ADDITIONAL EXPERIMENT SETTINGS

### C.1    DATASET

We used four widely adopted datasets to evaluate our proposed methods: CIFAR-10 (Krizhevsky & Hinton, 2009), CIFAR-100 (Krizhevsky & Hinton, 2009), Tiny-ImageNet (Le & Yang, 2015), and EMNIST (Cohen et al., 2017).

- CIFAR-10. CIFAR-10 consists of 60,000 color images with $32 \times 32$ pixels, evenly distributed across 10 classes.
- CIFAR-100. CIFAR-100 dataset consists of 60,000 $32 \times 32$ color images, evenly distributed across 100 classes.
- Tiny-ImageNet. Tiny-ImageNet dataset contains 110,000 images at $64 \times 64$ pixels across 200 classes and is a more challenging dataset with 500 training and 50 validation images per class.
- EMNIST (ByClass). The EMNIST dataset is an extension of the MNIST dataset, providing a more extensive set of handwritten character images. In our experiments, we use the byclass subset, which contains 814,255 images of size $28 \times 28$ pixels across 62 classes (10 digits (0-9) and 52 letters (uppercase A-Z and lowercase a-z)).

### C.2    IMPLEMENTATION

We conduct simulations of all clients and the server on a workstation equipped with an RTX 4090 GPU, a 2.6-GHz Intel(R) Xeon(R) W7-2475X CPU, and 125 GiB of RAM. The implementation of all methods is done using PyTorch.

## D    ADDITIONAL EXPERIMENTS

### D.1    ADDITIONAL EXPERIMENT WITH CIFAR-100/TINY-IMAGENET DATASET

We provide additional experiments over the CIFAR-100/Tiny-ImageNet datasets. Here, we compare MH-pFedHN and MH-pFedHNGD to the baselines on a small-scale setup of 10 clients. The results are presented in Tables 6. We show significant improvement using MH-pFedHN and MH-pFedHNGD on small-scale experiments in addition to the results presented in the main text.

### D.2    EXPERIMENTS WITH VISION TRANSFORMER USING CIFAR-100/TINY-IMAGENET DATASET

Additionally, we conducted further experiments using the Vision Transformer (ViT) model (ViT_Tiny_patch16_224) with 50 clients on CIFAR-100 and TinyImageNet datasets. The results are shown in Table 7. On the TinyImageNet dataset, the ViT model achieves accuracy improvements of up to 2.33% and 1.20% compared to LeNet under the non-IID_1 (44.68%) and non-IID_2 (59.50%) settings, respectively, which demonstrate that our approach is also applicable to large model scenarios.

Note that Vision Transformers have an extremely large number of parameters, making it highly challenging and time-consuming for a hypernetwork to generate Transformer models. Consequently, existing hypernetwork research has rarely explored Transformers. We will explore the use of hypernetworks for generating Vision Transformers in future work.

Table 6: Comparison under Homogeneous and Heterogeneous Model Settings (10 Clients)

| | Homogeneous Model | | | | Heterogeneous Model | | | |
| | CIFAR-100 | | Tiny-ImageNet | | CIFAR-100 | | Tiny-ImageNet | |
| Method | non-IID_1 | non-IID_2 | non-IID_1 | non-IID_2 | non-IID_1 | non-IID_2 | non-IID_1 | non-IID_2 |
|---|---|---|---|---|---|---|---|---|
| Local | 64.63 | 61.53 | 39.94 | 38.50 | 65.59 | 63.33 | 42.59 | 41.89 |
| FedAvg (McMahan et al., 2017) | 29.70 | 23.83 | 9.99 | 7.05 | 27.06 | 19.12 | 9.59 | 7.48 |
| pFedHN (Shamsian et al., 2021) | 63.68 | 61.91 | 41.05 | 38.73 | 60.57 | 60.76 | 39.44 | 40.24 |
| pFedLA (Ma et al., 2022) | 65.56 | 56.79 | 43.53 | 34.30 | 56.53 | 55.13 | 35.57 | 33.61 |
| FedGH (Yi et al., 2023a) | 65.84 | 63.34 | 41.34 | 39.43 | 63.04 | 61.65 | 36.48 | 31.84 |
| pFedLHN (Zhu et al., 2023) | 66.46 | 64.79 | 42.78 | 39.78 | 66.76 | 65.35 | 43.08 | 43.16 |
| PeFLL (Scott et al., 2024) | 52.82 | 48.31 | 37.59 | 33.48 | 61.14 | 58.57 | 41.20 | 41.38 |
| FedAKT (Liu et al., 2025) | **69.17** | 63.27 | 43.61 | 39.82 | 66.61 | 65.84 | 43.15 | 42.36 |
| MH-pFedHN (ours) | 68.12 | **64.81** | **44.38** | 40.59 | 67.02 | 66.46 | 44.80 | **45.14** |
| MH-pFedHNGD (ours) | 68.39 | **64.98** | **44.49** | **41.62** | 67.10 | 67.41 | 45.51 | 44.28 |

Table 7: Experiments with the ViT Model, where the left side is non-IID_1 and the right is non-IID_2.

| Datasets | CIFAR-100 | | Tiny-ImageNet | |
|---|---|---|---|---|
| ViT | 58.16 | 70.32 | 44.68 | 59.50 |

Table 8: MH-pFedHN and MH-pFedHNGD experiments with 500 clients, where green represents homogeneous models and red represents heterogeneous models.

| Data_Distribution | non-IID_1 | | | non-IID_2 | | |
| # Datasets | CIFAR-100 | Tiny-ImageNet | EMNIST | CIFAR-100 | Tiny-ImageNet | EMNIST |
|---|---|---|---|---|---|---|
| MH-pFedHN | 56.71 | 32.00 | 97.49 | 79.29 | 62.20 | 98.82 |
| MH-pFedHNGD | 57.11 | 35.96 | 97.53 | 80.21 | 66.19 | 98.99 |
| MH-pFedHN | 30.31 | 15.31 | 93.12 | 67.97 | 52.07 | 95.95 |
| MH-pFedHNGD | 33.08 | 16.07 | 97.30 | 68.69 | 53.26 | 98.07 |

Table 9: Test accuracy (%) over 10, 50, and 100 clients on CIFAR-10 under Homogeneous and Heterogeneous models with Heterogeneous data.

| | Homogeneous Model | | | | | | Heterogeneous Model | | | | | |
| | non-IID_1 | | | non-IID_2 | | | non-IID_1 | | | non-IID_2 | | |
| Method | 10 | 50 | 100 | 10 | 50 | 100 | 10 | 50 | 100 | 10 | 50 | 100 |
|---|---|---|---|---|---|---|---|---|---|---|---|---|
| Local | 92.71 | 89.38 | 87.87 | 83.04 | 82.77 | 79.53 | 93.23 | 84.98 | 80.26 | 81.97 | 80.58 | 75.09 |
| FedAvg | 50.15 | 51.39 | 58.28 | 64.23 | 64.77 | 61.65 | 54.59 | 65.51 | 57.28 | 73.61 | 56.75 | 51.65 |
| pFedHN | 92.60 | 91.66 | 90.87 | 82.67 | 85.49 | 81.78 | 92.83 | 88.11 | 87.00 | 81.12 | 80.96 | 77.75 |
| pFedLA | 85.12 | 83.86 | 81.71 | 77.49 | 84.06 | 77.14 | 93.26 | 73.27 | 65.38 | 77.67 | 81.00 | 75.55 |
| FedGH | 92.90 | 88.24 | 85.39 | 84.19 | 73.55 | 72.82 | 92.56 | 86.04 | 83.36 | 83.01 | 70.39 | 70.26 |
| pFedLHN | 94.16 | 91.09 | 90.71 | 82.49 | 86.10 | 85.48 | 93.67 | 89.33 | 87.26 | 83.79 | 84.21 | 79.82 |
| PeFLL | 89.21 | 86.62 | 86.08 | 75.59 | 78.25 | 78.76 | 92.21 | 89.09 | 88.20 | 80.85 | 82.49 | 80.65 |
| FedAKT | 93.46 | 91.14 | 90.31 | 83.88 | 85.96 | 82.58 | 93.58 | 88.59 | 85.62 | 82.82 | 83.05 | 78.69 |
| MH-pFedHN (ours) | **94.24** | 91.69 | 91.23 | 83.72 | **87.13** | 85.79 | **93.98** | 89.74 | 88.53 | **83.87** | 84.11 | 80.53 |
| MH-pFedHNGD (ours) | 93.56 | **91.75** | **92.12** | 82.92 | **87.50** | **86.25** | 93.81 | **90.26** | **88.73** | 83.83 | **85.52** | **80.88** |

## D.3 EXPERIMENTS WITH SCALABILITY

We also present results for MH-pFedHN and MH-pFedHNGD on 500 clients to verify that our approach scales effectively to large-scale scenarios. Results are shown in Table 8, which shows the scalability of our methods.

Table 10: MH-pFedHN and MH-pFedHNGD experiments with different participation ratios on 50 clients, where green represents homogeneous models and red represents heterogeneous models.

| Methods | CIFAR-100 | | | |
| | MH-pFedHN | | MH-pFedHNGD | |
| RATIO | non-IID_1 | non-IID_2 | non-IID_1 | non-IID_2 |
|---|---|---|---|---|
| C = 20% | 64.62 | 77.61 | 63.89 | 78.30 |
| C = 40% | 64.73 | 78.49 | 65.52 | 79.15 |
| C = 60% | 64.58 | 78.21 | 67.02 | 78.84 |
| C = 80% | 65.00 | 78.33 | 67.44 | 79.90 |
| C = 100% | 64.69 | 77.93 | 68.30 | 80.07 |
| C = 20% | 51.82 | 69.92 | 50.61 | 70.19 |
| C = 40% | 54.17 | 72.17 | 57.02 | 73.55 |
| C = 60% | 55.78 | 74.98 | 58.26 | 75.59 |
| C = 80% | 56.14 | 74.11 | 58.19 | 76.10 |
| C = 100% | 57.09 | 75.40 | 60.11 | 76.76 |

Table 11: Experiment with different architectures of global model, where the model structure before $\sim$ represents the personalized client model, and the one after $\sim$ represents the global model.

| Data_Distribution | CIFAR-100 | | | | | |
| | non-IID_1 | | | non-IID_2 | | |
| # Clients | 50 | 100 | 200 | 50 | 100 | 200 |
|---|---|---|---|---|---|---|
| VGGNet-8~VGGNet-8 | 63.24 | 50.67 | 44.18 | 77.88 | 77.27 | 77.54 |
| ResNet-10~ResNet-10 | 61.33 | 54.66 | 44.17 | 73.43 | 76.06 | 77.52 |
| MLP~MLP | 53.09 | 49.50 | 44.95 | 70.15 | 74.52 | 75.05 |
| VGGNet-8/ResNet-10/MLP~VGGNet8 | 59.95 | 51.01 | 44.49 | 74.39 | 77.14 | 75.97 |
| VGGNet-8/ResNet-10/MLP~ResNet-10 | 57.50 | 51.12 | 44.59 | 74.62 | 77.02 | 76.65 |
| VGGNet-8/ResNet-10/MLP~MLP | 58.33 | 49.17 | 43.94 | 74.67 | 76.58 | 76.36 |

## D.4 EXPERIMENTS WITH CIFAR-10 DATASET

We provide additional experiments over the CIFAR-10 dataset. Table 9 shows that in both non-IID settings, MH-pFedHN and MH-pFedHNGD achieve superior accuracy compared to baseline methods, highlighting their effectiveness in handling data heterogeneity.

## D.5 EXPERIMENTS WITH DIFFERENT CLIENT PARTICIPATION RATIOS

Here, we investigate the performance of MH-pFedHN and MH-pFedHNGD under partial client participation settings. The experiments are conducted on the CIFAR-100 dataset with 50 clients, where the client participation ratios are set to {0.2, 0.4, 0.6, 0.8}. The experimental results are shown in Table 10. It can be observed that our method is robust to pFL settings with varying client participation ratios. As the participation ratio increases, the training becomes more effective, leading to better performance.

Moreover, when the participation ratio is low, we find that the performance of MH-pFedHNGD is inferior to that of MH-pFedHN in the non-IID_1 setting. This is because non-IID_1 is more heterogeneous than non-IID_2. In this case, the global model can only represent the shared knowledge of a highly heterogeneous subset of clients, rather than the global shared knowledge, and thus fails to provide effective knowledge distillation guidance.

## D.6 EXPERIMENTS WITH DIFFERENT ARCHITECTURES OF GLOBAL MODEL

Table 11 shows the results of experiments with different architectures of the global model, which are similar to those in Table 5 when using a simple LeNet as the global model. This demonstrates the robustness of our method, as well as the effectiveness of our lightweight LeNet-based global model.

Table 12: The experiments exploring the impact of sharing the head on the performance of MH-pFedHN and MH-pFedHNGD, where green indicates shared heads and red indicates non-shared heads.

| Data_Distribution | CIFAR-100 | | | | | |
| | non-IID_1 | | | non-IID_2 | | |
| # Clients | 50 | 100 | 200 | 50 | 100 | 200 |
|---|---|---|---|---|---|---|
| MH-pFedHN | 55.92 | 51.26 | 45.56 | 73.92 | 76.65 | 76.57 |
| MH-pFedHNGD | 58.22 | 53.78 | 49.83 | 75.13 | 76.89 | 77.09 |
| MH-pFedHN | 57.84 | 51.75 | 47.03 | 74.53 | 76.73 | 76.74 |
| MH-pFedHNGD | 58.46 | 55.43 | 50.20 | 75.23 | 77.71 | 77.28 |

Table 13: Impact of redundant parameters on accuracy (%), where the left side is non-IID_1 and the right is non-IID_2 in the last row.

| | CIFAR-100 | | | | |
| Number of Embedding Vectors | 79 | 80 | 81 | 82 | 83 |
|---|---|---|---|---|---|
| Discarded | 2832 | 2832+3072 | 2832+2*3072 | 2832+3*3072 | 2832+4*3072 |
| MH-pFedHN | 64.69   77.93 | 64.87   78.58 | 65.13   78.96 | 64.46   78.17 | 65.30   78.55 |

### D.7 EXPERIMENTS WITH REDUNDANT PARAMETERS

In our method, when the number of model parameters required by client $i$ is $K_i$, we assign $\tau_i = \lceil K_i/N \rceil$ embedding vectors to client $i$. Since each embedding vector is fed into the hypernetwork to output $1 \times N$ parameters, the hypernetwork generates $\tau_i \times N \geq K_i$ parameters for client $i$ in total. However, only the first $K_i$ parameters are sent to and used by the client; the redundant parameters beyond $K_i$ are directly discarded and are not involved in the client's local training or inference process. This design ensures that each client receives exactly the number of parameters it actually needs, and the redundant parameters have no impact on the overall process.

We conducted experiments on the CIFAR-100 dataset with 50 clients under a homogeneous setting to verify the impact of discarding redundant parameters. In this setting, the server creates 79 embedding vectors for each client and discards the last 2832 out of the 3072 parameters generated by the last embedding vector. For further experiments, we increased the number of embedding vectors per client from 79 to 83, so the number of discarded parameters would further increase. The experimental results are shown in Table 13, and it can be found that discarding these redundant parameters does not have a negative impact.

### D.8 EXPERIMENTS WITH SHARED HEADS FOR HETEROGENEOUS MODELS WITH SIMILAR PARAMETER SIZES

To investigate the impact of sharing the head among clients with the same number of model parameters but different structures on the performance of MH-pFedHN and MH-pFedHNGD, we adjusted the VGGNet and MLP models to match the number of parameters in the LeNet model. Consequently, the server created the same number of embedding vectors for these clients and treated their models as homogeneous, enabling all clients to share the same head. The experimental results are shown in the green section of Table 12. Additionally, we processed the clients using these three models in a manner where the number of embedding vectors varied, meaning that only clients using the same model could share the same head. The results of this setup are presented in the red section.

According to the experimental results, using different heads for models with the same number of parameters but different architectures performs slightly better than using the same head for different architectures. However, since the server generally cannot access the specific model architecture information of the clients, our method is sufficiently robust; under the premise of protecting model

Table 14: Performance comparison of our methods with FedProto and FedTGP on CIFAR-100 Dataset under heterogeneous settings with 50, 100, and 200 Clients.

| Data_Distribution | CIFAR-100 | | | | | |
| | non-IID_1 | | | non-IID_2 | | |
| # Clients | 50 | 100 | 200 | 50 | 100 | 200 |
| --- | --- | --- | --- | --- | --- | --- |
| FedProto | 31.21 | 30.73 | 25.49 | 55.05 | 56.85 | 50.46 |
| FedTGP | 50.31 | 41.68 | 28.39 | 69.54 | 71.62 | 67.64 |
| MH-pFedHN | 57.09 | 50.35 | 42.98 | 75.40 | 77.24 | 75.38 |
| MH-pFedHNGD | 60.11 | 52.14 | 43.41.20 | 76.76 | 77.03 | 76.46 |

Table 15: Communication and computation overhead under non-IID_1, the results are similar in two non-IID settings. Green indicates MH-pFedHN, red indicates MH-pFedHNGD.

| Algorithm | Overall Computation (mins) | Overall Communication (MB) |
| --- | --- | --- |
| homogeneous | 77.01 | 1.83 |
| heterogeneous | 303.8 | 3.00 |
| homogeneous | 133.7 | 3.66 |
| heterogeneous | 407.8 | 4.83 |

Table 16: Communication rounds and time (seconds) to reach the same accuracy, where the left side is the communication round number and the time (seconds).

| Up to | pFedHN | | pFedLHN | | MH-pFedHN | | MH-pFedHNGD | |
| --- | --- | --- | --- | --- | --- | --- | --- | --- |
| 20% | 8 | 83.46 | 14 | 162.53 | 16 | 159.26 | 14 | 294.98 |
| 40% | 20 | 178.37 | 21 | 239.57 | 21 | 209.19 | 19 | 393.69 |
| 60% | 231 | 2224.84 | 105 | 1183.96 | 39 | 434.98 | 33 | 672.73 |

structure privacy, it achieves a reasonable balance between personalized performance and privacy protection.

## D.9 EXPERIMENTS WITH MORE METHODS FOR MODEL HETEROGENEITY

To further evaluate the performance of our methods in heterogeneous scenarios, we included two representative FL methods for comparison: FedProto (Tan et al., 2022) and FedTGP (Zhang et al., 2024b). FedProto and FedTGP are prototype-centered FL methods, which utilize class prototypes to handle data heterogeneity and model heterogeneity.

We conducted experiments on CIFAR-100 dataset under heterogeneous settings with 50, 100, and 200 clients, comparing our methods with FedProto and FedTGP. The results are shown in the Table 14. Notably, our methods demonstrate greater effectiveness when faced with the more extreme heterogeneity in our settings. Additionally, we conducted further experiments using the HtFE2 heterogeneous setting in FedTGP on CIFAR-100 with 50 clients. The results are 63.09 (non-IID_1) and 76.96 (non-IID_2), which further confirms that our methods can be applied to more heterogeneous settings.

## D.10 EXPERIMENTS WITH COMMUNICATION AND COMPUTATION OVERHEAD

Table 15 shows the communication and computation overhead of MH-pFedHN and MH-pFedHNGD for 50 clients with 500 rounds on CIFAR-100. The computation overhead of MH-pFedHNGD is 2.0 and 1.61 times that of MH-pFedHN in homogeneous and heterogeneous settings, respectively, which is consistent with our experimental setup. The computation overhead for MH-pFedHNGD is higher than that of MH-pFedHN by 73% and 34% in homogeneous and heterogeneous settings, respectively. This indicates that our lightweight global model introduces only minimal additional time complexity while significantly enhancing the generalization ability of the hypernetwork and improving overall performance, especially for heterogeneous settings.

Table 17: We used CIFAR-100 to evaluate the impact on MH-pFedHN and MH-pFedHNGD of only accepting the first 30% weight update with the largest absolute value, where green and red correspond to the homogeneous and the heterogeneous models, respectively.

| Data_Distribution | CIFAR-100 | | | | | |
| | non-IID_1 | | | non-IID_2 | | |
| # Clients | 50 | 100 | 200 | 50 | 100 | 200 |
| MH-pFedHN | 63.91 | 63.14 | 59.25 | 78.25 | 80.79 | 81.04 |
| Top 30% | 64.35 | 62.97 | 58.39 | 78.78 | 80.91 | 81.10 |
| MH-pFedHNGD | 68.27 | 63.58 | 61.19 | 80.18 | 82.31 | 82.19 |
| Top 30% | 67.79 | 63.96 | 61.56 | 80.22 | 83.00 | 82.92 |
| MH-pFedHN | 57.18 | 50.35 | 40.83 | 74.84 | 76.78 | 74.71 |
| Top 30% | 57.35 | 47.61 | 40.68 | 75.33 | 77.11 | 75.35 |
| MH-pFedHNGD | 60.11 | 51.57 | 43.41 | 76.76 | 77.03 | 76.46 |
| Top 30% | 59.39 | 50.80 | 43.55 | 75.97 | 77.55 | 75.68 |

Table 18: Experiment with extreme personalized setting on CIFAR-100.

| | | CIFAR-100 | | | | | |
| | Data_Distribution | non-IID_1 | | | non-IID_2 | | |
| | # Clients | 50 | 100 | 200 | 50 | 100 | 200 |
| Homogeneous model | MH-pFedHN | 56.96 | 51.62 | 44.38 | 75.59 | 77.71 | 78.53 |
| | MH-pFedHNGD | 59.67 | 56.20 | 49.23 | 76.58 | 79.09 | 79.99 |
| Heterogeneous model | MH-pFedHN | 54.37 | 46.00 | 38.80 | 74.49 | 76.32 | 75.38 |
| | MH-pFedHNGD | 58.71 | 50.07 | 41.79 | 77.23 | 78.16 | 76.59 |

Additionally, in a homogeneous environment, we measured the communication rounds and time (in seconds) required by each method to achieve accuracy levels of approximately 20%, 40%, and 60%. The results are presented in Table 16. We observe that our method achieves higher accuracy with fewer communication rounds and reduced communication time.

## D.11 EXPERIMENTS WITH COMMUNICATION EFFICIENCY USING WEIGHT PRUNING

Due to the introduction of a global model in MH-pFedHNGD, although its number of parameters is the same as the minimum number of parameters required by the client, it still increases communication overhead. We considered weight pruning, which involves pruning the weight updates of personalized models uploaded by the client to the server and only uploading the top 30% of updates with the largest absolute values in each round of communication. We also applied this pruning method to MH-pFedHN, with the experimental results presented in Table 17. Additionally, specific parameter details can be found in Appendix G.

Unexpectedly, weight pruning did not significantly decrease accuracy. The model's performance is comparable to that when no pruning is applied. This indicates that the hypernetwork of our methods can still learn efficiently and maintain high performance even with less information, and has great potential for practical use, effectively retaining key information and ensuring that model accuracy is not compromised.

## D.12 EXPERIMENTS WITH EXTREME PERSONALIZED SETTING

Here, we fully consider the generation of personalized parameters for clients, which allows clients to choose to retain parameters for certain layers locally without participating in federated training under the extremely personalized setting. This greatly satisfies the personalized needs of the clients.

Following the settings in Section 4.2 and 4.3, in the experiments with homogeneous models, all clients use a LeNet-style model. We have set up four modes, 1/4 of the clients are required to generate parameters for the input layer and hidden layers, while retaining the parameters for the

Table 19: MH-pFedHN in resource-constrained experiments. Test accuracy over 50, 100, and 200 clients on the CIFAR-100.

|  |  | CIFAR-100 | | | | | |
|  |  | non-IID_1 | | | non-IID_2 | | |
| Data_Distribution |  | 50 | 100 | 200 | 50 | 100 | 200 |
| # Clients |  |  |  |  |  |  |  |
| Homogeneous model | MH-pFedHN | 53.61 | 45.30 | 40.93 | 73.91 | 77.37 | 77.78 |
| Heterogeneous model | MH-pFedHN | 53.28 | 46.08 | 37.67 | 75.10 | 76.72 | 76.01 |

Table 20: MH-pFedHNGD in resource-constrained experiments, where green represents homogeneous models and red represents heterogeneous models.

| Methods | CIFAR-100 | | | | | |
|  | non-IID_1 | | | non-IID_2 | | |
| # Clients | 50 | 100 | 200 | 50 | 100 | 200 |
| C = 0% | 64.69 | 63.32 | 60.11 | 77.93 | 80.93 | 81.60 |
| C = 20% | 64.82 | 63.41 | 60.42 | 78.87 | 80.91 | 81.93 |
| C = 40% | 65.17 | 63.07 | 60.34 | 78.63 | 81.85 | 82.48 |
| C = 60% | 66.48 | 62.60 | 61.05 | 79.22 | 82.05 | 82.59 |
| C = 80% | 66.50 | 63.38 | 61.33 | 79.43 | 82.77 | 82.79 |
| C = 100% | 68.30 | 63.97 | 61.59 | 80.07 | 82.51 | 82.54 |
| C = 0% | 57.09 | 50.53 | 42.98 | 75.40 | 77.24 | 75.38 |
| C = 20% | 57.48 | 50.23 | 42.55 | 75.90 | 76.95 | 75.21 |
| C = 40% | 58.48 | 49.17 | 41.11 | 74.90 | 77.03 | 75.07 |
| C = 60% | 58.06 | 51.42 | 42.64 | 75.56 | 76.79 | 74.90 |
| C = 80% | 58.79 | 50.63 | 37.38 | 76.44 | 77.51 | 76.22 |
| C = 100% | 60.11 | 52.14 | 43.41 | 76.76 | 77.03 | 76.46 |

classification layer locally; 1/4 of the clients are required to generate parameters for the hidden layers, retaining the parameters for the input and classification layers locally; 1/4 of the clients are required to generate parameters for the hidden and classification layers, retaining the parameters for the input layer locally; and the final 1/4 of the clients are required to generate parameters for all layers. In the experiments with heterogeneous models, the setup for the VGG model is similar to that of the LeNet-style model. For the residual networks, we configure some clients to generate only the parameters for the convolutional layers, retaining the parameters for the batch normalization layers and classification layers locally, while other clients are required to generate all parameters. Therefore, in the experiments with heterogeneous models, there are 14 different modes, with the number of clients in each mode being 1/14 of the total.

The experimental results, as shown in Table 18, indicate that when client personalization needs are met, the accuracy decreases to varying degrees. Therefore, a trade-off between personalization and performance is necessary.

## D.13 Experiments under Resource Constraint Setting

Here, we first investigate the performance of MH-pFedHN under resource-constrained conditions. In resource-constrained environments, such as on-edge devices or mobile platforms, clients often face limitations in computational power, memory, and storage capacity, significantly impacting their ability to effectively train deep learning models. Training a complete model with all layers requires substantial computational resources, which may be beyond the capabilities of these clients. Additionally, deep neural networks consume significant amounts of memory, making it challenging to load an entire model, especially for clients with limited memory capacity. Given these constraints, clients can only feasibly train a subset of the model, typically focusing on the shallower layers. This approach reduces the number of parameters, thereby lowering the memory and computational requirements.

To simulate such a scenario, we adhered to the configurations outlined in Sections 4.2 and 4.3 of our study. In the experiments with homogeneous models, all clients use a LeNet-style model. We

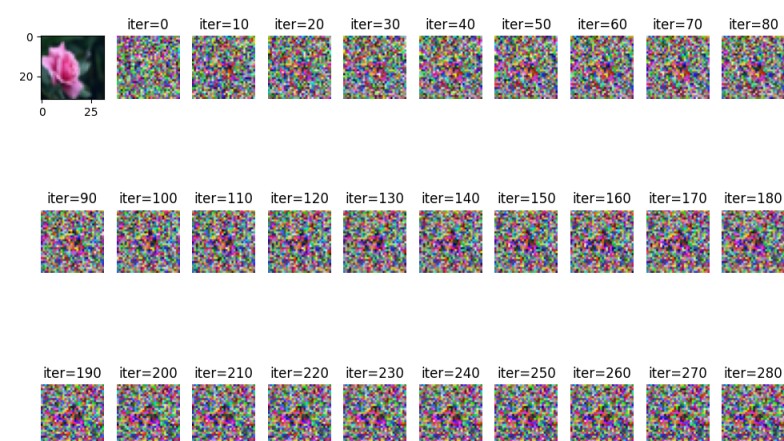

Figure 6: iDLG Gradient Inversion Attack using MH-pFedHNGD.

have set up three modes. In the first setting, we froze the parameters of the second fully connected layer and the classification layer, simulating the most severe resource constraints. In the second setting, we froze only the parameters of the classification layer. In the third setting, we allowed clients without resource limitations to train the entire model. These three settings were evenly distributed among the clients. In experiments with heterogeneous models, the VGG model's configuration was similar to that of the LeNet-style model. For residual networks, we established two scenarios: in the first scenario, we froze the parameters of the last 20% of the layers to mimic resource-constrained situations. In the second scenario, clients had no resource limitations and could therefore train the entire model. Likewise, these 12 different configurations were also evenly distributed among the clients. This design allows us to assess the performance variations in different resource availability situations, providing insights into how model architecture and training strategies influence overall effectiveness in heterogeneous client environments.

The experimental results are summarized in Table 19. We can observe that under resource-constrained conditions, the accuracy decreases to varying degrees across the different configurations. In particular, in the case of the non-IID_1 distribution, the decrease in accuracy is more pronounced. This can be attributed to the greater heterogeneity of the non-IID_1 scenario, where, on average, each client receives a smaller amount of data.

Next, we conducted an in-depth study of the performance of the MH-pFedHNGD algorithm in resource-constrained environments. In such scenarios, only a portion of clients can deploy the global model. We set the deployment ratios to 20%, 40%, 60%, and 80% of clients, respectively, to systematically evaluate the impact of different deployment scales on overall performance. As shown in Table 20, the analysis of the experimental data reveals that when the proportion of clients deploying the global model reaches 40%, there is a significant improvement in system performance. This indicates that in resource-limited environments, not all clients need to deploy the global model; as long as a certain proportion of clients can utilize the global model, the overall performance of the federated learning system can be effectively enhanced. This finding provides important guidance for optimizing resource allocation and improving system efficiency in practical applications.

# E   EXPERIMENTS WITH PRIVACY CONCERNS

## E.1   EXPERIMENTS WITH IDLG ATTACKS USING MH-PFEDHNGD

Figure 6 shows the results of iDLG (Zhao et al., 2020) using MH-pFedHNGD, which exhibits that our methods could still preserve the data security for clients even using the plug-in component global model.

Table 21: Accuracy of MH-pFedHN and MH-pFedHNGD under varying Laplacian noise levels, where the left side is non-IID_1 and the right is non-IID_2.

| | CIFAR-100 | | | | | | | | | | |
|---|---|---|---|---|---|---|---|---|---|---|---|
| # Laplace noise | 0.00625 | | 0.01 | | 0.0125 | | 0.01667 | | 0.02 | | 0.333 | |
| MH-pFedHN | 59.07 | 74.79 | 58.75 | 75.23 | 58.02 | 74.91 | 56.37 | 76.31 | 54.23 | 71.87 | 45.70 | 64.48 |
| MH-pFedHNGD | 59.71 | 75.92 | 60.33 | 75.82 | 59.40 | 75.08 | 57.17 | 72.44 | 53.59 | 71.15 | 45.54 | 63.76 |

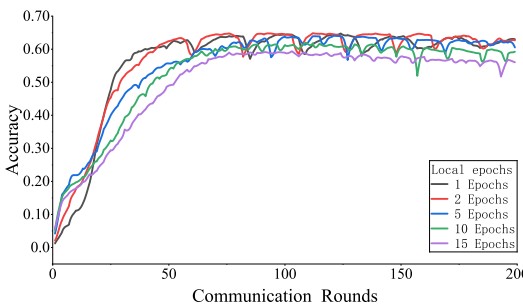

Figure 7: Experiment with the number of local epochs on the CIFAR-100 dataset using MH-pFedHN.

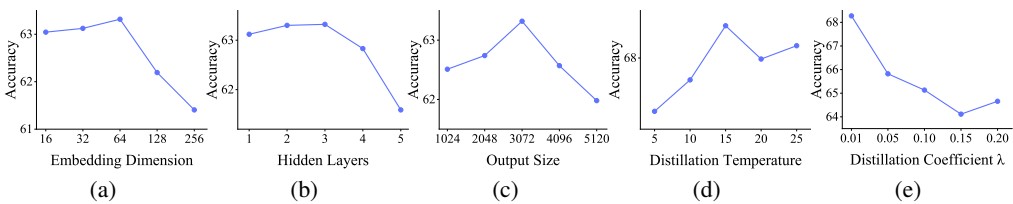

|  (a)  |  (b)  |  (c)  |  (d)  |  (e)  |

Figure 8: MH-pFedHN test results on CIFAR-100 showing the effect of (a) the dimension of client embedding vector, (b) the number of hypernetwork's hidden layers, and (c) the output size of hypernetworks, (d) the distillation temperature with MH-pFedHNGD, and (e) the balancing factor on distillation loss with MH-pFedHNGD.

### E.2    EXPERIMENTS WITH DIFFERENTIAL PRIVACY

We conducted differential privacy (Abadi et al., 2016) experiments for MH-pFedHN and MH-pFedHNGD on CIFAR-100 dataset with a setup of 50 clients. Specifically, for MH-pFedHN, Laplacian noise is added to the personalized weight updates sent from each client to the server, whereas for MH-pFedHNGD, Laplacian noise is applied not only to the personalized weight updates but also to the global model weight updates transmitted to the server. The results, presented in Table 21, show that as the Laplacian noise increases, the performance of our methods slightly degrades; however, both methods still maintain high accuracy. This indicates the effective integration of our approaches with differential privacy for robust privacy protection.

## F    EXPERIMENTS WITH HYPERPARAMETER CHOICES

### F.1    EXPERIMENTS WITH NUMBER OF LOCAL EPOCHS

Here, we set local epochs to {1,2,5,10,15}. Figure 7 shows the test accuracy throughout training over 200 rounds. Results indicate that MH-pFedHN is relatively robust to the choice of local optimization steps.

## F.2   EXPERIMENTS WITH CLIENT EMBEDDING DIMENSION

We investigate the effect of the dimension of the embedding vector on the performance of MH-pFedHN. Specifically, we run an ablation study on a set of different embedding dimensions {16, 32, 64, 128, 256}. The results are presented in Figure 8(a). We show MH-pFedHN robustness to the dimension of the client embedding vector; hence, we fix the embedding dimension through all experiments to 64.

## F.3   EXPERIMENTS WITH HYPERNETWORK CAPACITY

Here, we inspect the effect of the Hypernetwork's capacity on the local network performance. We conducted an experiment in which we changed the depth of the Hypernetwork by stacking fully connected layers. We evaluate MH-pFedHN using different hidden layers {1, 2, 3, 4, 5}. Figure 8(b) presents the final test accuracy. MH-pFedHN achieves optimal performance with three hidden layers. We use three hidden layers of HN for all experiments in the main text.

## F.4   EXPERIMENTS WITH OUTPUT SIZE OF HYPERNETWORKS

We investigate how the output size of hypernetworks impacts the performance of local networks. In our experimentation, we adjusted the output size to see its effects. We evaluated MH-pFedHN in various output sizes {1024, 2048, 3072, 4096, 5120}. The findings presented in Figure 8(c) indicate that MH-pFedHN performs best with an output size of 3072, which we use for all subsequent experiments in the main text.

## F.5   EXPERIMENTS WITH TEMPERATURE FOR KNOWLEDGE DISTILLATION

We examine the influence of the temperature parameter, which is crucial in knowledge distillation, on the performance of the MH-pFedHNGD framework using the CIFAR-100 dataset. We systematically varied the temperature settings to investigate their impact on the efficiency of MH-pFedHNGD. The specific temperature values tested are {5, 10, 15, 20, 25}. The results, illustrated in Figure 8(d), demonstrate that MH-pFedHNGD achieves optimal performance with a temperature of 15 in the CIFAR-100 dataset under the non-IID_1 setting. This optimal temperature enhances the softmax distribution, thereby improving the transfer of knowledge from the global model to the private model within MH-pFedHNGD. Therefore, we utilize this temperature setting for all experiments conducted on the CIFAR-100 dataset.

## F.6   EXPERIMENTS WITH BALANCING FACTOR ON DISTILLATION LOSS

We examine the influence of the balancing factor in the loss function, which is critical in knowledge distillation, on the performance of the MH-pFedHNGD framework using the CIFAR-100 dataset. We systematically varied the ratio between the true loss and the distillation loss to investigate their impact on the efficiency of MH-pFedHNGD. The specific balancing factor values tested were {0.01, 0.05, 0.1, 0.15, 0.2}. The results, illustrated in Figure 8(e), demonstrate that MH-pFedHNGD achieves optimal performance with a balancing factor of 0.01 in the CIFAR-100 dataset under the non-IID_1 setting. This optimal ratio facilitates an effective trade-off between accurately modeling the true data distribution and leveraging the global model's knowledge, thereby improving the overall learning process. Consequently, we utilize this balancing factor setting for all experiments conducted on the CIFAR-100 dataset.

## F.7   DISCUSSIONS ABOUT PRESERVING AND ENHANCING THE PERSONALIZED ABILITY

Based on the discussions of Appendix F.5 and Appendix F.6, by adjusting the distillation temperature, we control the smoothness of the soft labels during knowledge distillation, which affects the extent to which the local model absorbs knowledge from the global model. By regulating the distillation loss ratio, we balance the weight between the distillation loss and the local task loss, ensuring that the local model can effectively leverage global knowledge while preserving its personalized capabilities, which serves as one of the novelties of our second approach.

Table 22: LeNet-style model structure.

| Layer | Shape | Nonlinearity |
|---|---|---|
| Conv1 | $3 \times 3 \times 3 \times 16$ | ReLU |
| MaxPool | $2 \times 2$ | - |
| Conv2 | $16 \times 3 \times 3 \times 32$ | ReLU |
| MaxPool | $2 \times 2$ | Flatten |
| FC1 | $2048 \times 108$ | ReLU |
| FC2 | $108 \times 64$ | ReLU |
| FC3 | $64 \times 100$ | None |

Table 23: MLP model structure.

| Layer | Shape | Nonlinearity |
|---|---|---|
| FC1 | $3072 \times 128$ | ReLU |
| FC2 | $128 \times 64$ | ReLU |
| FC3 | $64 \times 100$ | None |

Table 24: Simplified VGG8 model structure.

| Layer | Shape | Nonlinearity |
|---|---|---|
| Conv1 | $3 \times 3 \times 3 \times 16$ | ReLU |
| Conv2 | $16 \times 3 \times 3 \times 16$ | ReLU |
| MaxPool | $2 \times 2$ | - |
| Conv3 | $16 \times 3 \times 3 \times 32$ | ReLU |
| Conv4 | $32 \times 3 \times 3 \times 32$ | ReLU |
| MaxPool | $2 \times 2$ | - |
| Conv5 | $32 \times 3 \times 3 \times 64$ | ReLU |
| Conv6 | $64 \times 3 \times 3 \times 64$ | ReLU |
| MaxPool | $2 \times 2$ | Flatten |
| Linear1 | $1024 \times 180$ | ReLU |
| Linear2 | $180 \times 64$ | ReLU |
| Linear3 | $64 \times 100$ | None |

Table 25: Structure of three Residual network models.

| Group Name | Output Size | 10-layer ResNet | 12-layer ResNet | 18-layer ResNet |
|---|---|---|---|---|
| Conv1 | $32 \times 32$ | $[3 \times 3, 16]$ | $[3 \times 3, 16]$ | $[3 \times 3, 16]$ |
| Conv2 | $32 \times 32$ | $\begin{bmatrix} 3 \times 3, 16 \\ 3 \times 3, 16 \end{bmatrix} \times 3$ | $\begin{bmatrix} 3 \times 3, 16 \\ 3 \times 3, 16 \end{bmatrix} \times 1$ | $\begin{bmatrix} 3 \times 3, 16 \\ 3 \times 3, 16 \end{bmatrix} \times 6$ |
| Conv3 | $16 \times 16$ | $\begin{bmatrix} 3 \times 3, 32 \\ 3 \times 3, 32 \end{bmatrix} \times 3$ | $\begin{bmatrix} 3 \times 3, 32 \\ 3 \times 3, 32 \end{bmatrix} \times 5$ | $\begin{bmatrix} 3 \times 3, 32 \\ 3 \times 3, 32 \end{bmatrix} \times 6$ |
| Conv4 | $8 \times 8$ | $\begin{bmatrix} 3 \times 3, 64 \\ 3 \times 3, 64 \end{bmatrix} \times 4$ | $\begin{bmatrix} 3 \times 3, 64 \\ 3 \times 3, 64 \end{bmatrix} \times 6$ | $\begin{bmatrix} 3 \times 3, 64 \\ 3 \times 3, 64 \end{bmatrix} \times 6$ |
| Avg-Pool | $1 \times 1$ | $[8 \times 8]$ | $[8 \times 8]$ | $[8 \times 8]$ |

Table 26: Comparison of model parameter sizes uploaded before and after weight pruning across different models in the CIFAR-100 task.

| Model | Params | Top 30% |
|---|---|---|
| LeNet-style Model | 0.915 M | 0.274M |
| Simplified VGG8 | 1.048 M | 0.314M |
| 10-layer ResNet | 1.347 M | 0.404M |
| 12-layer ResNet | 2.018 M | 0.605M |
| 18-layer ResNet | 2.179 M | 0.654M |

# G  MODEL ARCHITECTURES AND PARAMETERS

Here, we present all the model architectures and the parameters used in the CIFAR-100 experiments. Table 22 is a LeNet-style model, Table 23 is an MLP model, Table 24 is a simplified VGG model (8 layers), Table 25 is three residual networks, and Table 26 lists the parameters of all models, as well as those that need to be uploaded after pruning.

## H    LIMITATION

### H.1    REAL-WORLD APPLICATION SCENARIOS

Although our current work is evaluated primarily in a simulated environment, our method is highly suitable for practical real-world scenarios, especially those involving heterogeneous devices or models—such as collaborative learning across different healthcare institutions, IoT devices from various manufacturers, or personal mobile devices with diverse architectures.

In fact, hypernetwork-based federated learning methods similar to ours have already been considered for real-world applications. For example, as demonstrated by the work (Shin et al., 2024a), their hypernetwork-based weight generation framework was effectively applied to embedded networked sensor systems, showing the feasibility and effectiveness of such approaches in real-world heterogeneous federated learning applications, i.e., human activity recognition.

Therefore, we believe that our method can be readily extended and deployed in practical federated learning scenarios.

### H.2    POTENTIAL LIMITATIONS

One limitation of using hypernetworks is their dependency on server resources, as they require substantial computational power and storage capacity. This characteristic presents high demands on server infrastructure and is the inherent characteristic of hypernetwork architectures. Currently, our work primarily addresses the challenges posed by MH-pFedHNGD. In particular, the incorporation of a global model in MH-pFedHNGD increases both the system overhead on the client side and the communication overhead between clients and the server.

To alleviate these issues, we aim, in future work, to explore more efficient and convenient methods that can achieve the same objectives as integrating a global model, without imposing additional burdens on client-server communication or increasing system overhead on the client side.

Further, we note that switching from homogeneous models to heterogeneous models within the same settings can lead to a decline in accuracy, which represents a challenge that we aim to address in future research.

## I    THE USE OF LARGE LANGUAGE MODELS

We only use Large Language Models (LLMs) to aid or polish writing and check typos.

