# OpenReview forum: "Revisiting Hypernetwork in Model-Heterogeneous Personalized Federated Learning"
_ICLR.cc/2026/Conference — ICLR 2026 Conference Withdrawn Submission_

### Official Review · Reviewer_L8os · 2025-10-16

**Soundness:** 2
**Presentation:** 2
**Contribution:** 2
**Rating:** 2
**Confidence:** 4

**Summary:**

This paper introduces MH-pFedHN and its improved variant MH-pFedHNGD, two hypernetwork-based frameworks for model-heterogeneous personalized federated learning (pFL). The methods train a central hypernetwork to generate personalized client models with different architectures. MH-pFedHNGD further integrates a lightweight global model for knowledge distillation. The paper provides extensive experiments on multiple datasets show that both variants outperform existing baselines.

**Strengths:**

- The paper is easy to follow.
- The improved variant, MH-pFedHNGD, adds a lightweight global model for distillation, offering a conceptually simple yet effective way to improve performance.
- The proposed methods achieve strong empirical results, and the paper provides a comprehensive evaluation.

**Weaknesses:**

- While empirical results are good, my main concern is the novelty of the proposed methods. The core idea of using a hypernetwork to generate personalized models in federated learning has already been established by pFedHN (Shamsian et al., 2021), pFedLHN (Zhu et al., 2023), and many other works. The proposed HN framework provides minor structural adjustments. The improvement, MH-pFedHNGD, is a fairly straightforward combination of existing hypernetwork and distillation ideas seen in FedKD and FedDF. The integration is useful but conceptually incremental rather than novel.
- The paper does not provide a theoretical analysis or deeper empirical explanation for why the added global distillation step improves generalization, beyond empirical gains.
- The paper lacks ablations and justification for key design choices. For example, the effects of the shared-head mechanism and lightweight global model should be better analyzed.
- Please report standard errors or another measure of variability for the results to clarify the robustness and significance of the performance gains.

**Questions:**

- How much does the shared-head mechanism contribute compared to using separate heads?
- How does the proposed method’s computational and communication overhead compare to prior hypernetwork-based and standard pFL approaches?

---

### Official Review · Reviewer_a2Va · 2025-10-24

**Soundness:** 3
**Presentation:** 2
**Contribution:** 3
**Rating:** 6
**Confidence:** 2

**Summary:**

This paper introduces MH-pFedHN, a server-side hypernetwork that generates client-specific parameters for heterogeneous models using customized embedding vectors and shared heads grouped by similar parameter sizes. The key advantage of this formulation is that this lets the server produce weights for multiple architectures without learning their structure. The paper further adds MH-pFedHNGD, which introduces a lightweight global model generated by the same hypernetwork to (1) update the hypernetwork and embeddings using clients’ global updates and (2) distill into each client’s personalized model via a KL term. The experiments and ablations show consistent performance gains on pFL baselines in both homogeneous and heterogeneous settings.

**Strengths:**

- The customized embedding + shared head design gives a concrete mechanism to map parameter counts to weight slices which allows one-pass generation for multiple models while preserving model privacy.
- The MH-pFedHNGD extension is relatively easy to implement as a small global model improves the hypernetwork via an extra update path and guides personalization with a KL term.
- Generalization studies show strong performance on unseen clients and some robustness to new architectures, showing a viable method on providing consolidated knowledge to multiple models.
- The experiments are comprehensive and shows consistent improvements.

**Weaknesses:**

1. The paper asserts that head sharing by parameter count should enable knowledge transfer across different architectures but, this claim is not quantified.
2. I'm still not clear on the theoretical advantages of global distillation during MH-pFedHNGD. When exactly does  distillation help or hurt (e.g., mismatch between global and local capacity) beyond brief improvements?
3. It's unclear what would happen when two models have equal parameter counts but very different layer architectures. Does the shared head saturate or overfit certain layers?

**Questions:**

All my questions are included in the weaknesses above.

---

### Official Review · Reviewer_GPy1 · 2025-10-26

**Soundness:** 2
**Presentation:** 2
**Contribution:** 1
**Rating:** 2
**Confidence:** 5

**Summary:**

This paper proposes a hypernetwork-based personalized federated learning framework, MH-pFedHN, to address generalization issues in personalized federated learning (pFL). By leveraging a shared hypernetwork and a shared global model, the method achieves reasonable performance without relying on external data or revealing client model architectures. However, the paper does not sufficiently discuss the existing challenges associated with model heterogeneity, and the two proposed algorithms are highly similar, which reduces their distinctiveness. Overall, the contribution appears incremental and lacks sufficient novelty or depth to advance the current state of the field.

**Strengths:**

-Unlike many existing studies that focus primarily on data heterogeneity in pFL, this work addresses the more practical and challenging problem of model heterogeneity in federated learning. Tackling this aspect enhances the real-world applicability and robustness of FL systems.

-The proposed framework leverages a hypernetwork with global embedding vectors to facilitate knowledge distillation. By exploiting the representational capability of the hypernetwork, the method effectively transfers knowledge without requiring external data, additional knowledge sources, or pretrained models.

**Weaknesses:**

-The paper does not clearly clarify the core problem it aims to solve. Although the authors list several shortcomings of existing algorithms in introduction part and claim to address them all, these issues are mostly technical refinements, which makes the contribution appear incremental. It remains unclear why prior works employed techniques such as model decoupling, partial training, or distillation to handle model heterogeneity, what specific aspects of heterogeneity were effectively addressed, and which remain unresolved?

-The reason for using the same embedding vectors for clients with a similar number of parameters is unclear. The number of parameters alone may not serve as a meaningful or discriminative feature for the hypernetwork to learn from, since models with comparable parameter counts can differ substantially in architecture and functional behavior. Even subtle variations in key architectural components can lead to markedly different outcomes. A similar concern applies to the design of the shared head.

-The distinction between MH-pFedHN and MH-pFedHNGD appears rather limited. The latter seems to be a straightforward enhancement of the former rather than a significantly different algorithm. MH-pFedHNGD could be viewed as an improved variant of MH-pFedHN, suggesting that the two should not be presented as separate core contributions. This overlap weakens the overall novelty and dilutes the clarity of the paper’s contribution structure.

**Questions:**

-It remains unclear how the hypernetwork can generalize effectively to unseen clients with architectures and parameter numbers that differ from those encountered during training. The current framework does not appear to include a mechanism for learning or adapting suitable client embeddings. This omission raises concerns about the model’s ability to generalize across diverse client architectures and ultimately weakens the claimed contribution to improving generalization in pFL.

-It is unclear how clients are grouped within the proposed framework. Which component enables the proposed framework to infer or access information about each client’s model architecture? This raises a potential inconsistency with the paper’s claim that the method operates without disclosing client model information. If architectural details are implicitly required for grouping or embedding generation, this could contradict the stated privacy-preserving objective.

---

### Official Review · Reviewer_Fgew · 2025-10-31

**Soundness:** 2
**Presentation:** 2
**Contribution:** 2
**Rating:** 4
**Confidence:** 4

**Summary:**

This paper proposes **MH-pFedHN** (Multi-Head personalized Federated Hypernetwork), a hypernetwork-based approach for personalized federated learning that supports heterogeneous model architectures across clients. The key idea is to use multiple heads in the hypernetwork, where each head generates parameters for different model architectures. The paper also introduces **MH-pFedHNGD**, which incorporates knowledge distillation to improve performance. The method allows each client to have a customized model architecture while maintaining collaborative learning through a shared hypernetwork backbone.

**Strengths:**

- **Multi-head hypernetwork for heterogeneous architectures**: The idea of using multi-head hypernetwork to support heterogeneous model architectures in federated learning is straightforward and intuitive, addressing a practical problem in real-world federated learning scenarios where clients may have different computational capabilities.

- **Strong empirical improvements**: The proposed method demonstrates effectiveness with significant improvements compared to baseline methods in the experimental results, showing the potential of the approach.

- **Comprehensive experimental coverage**: The paper provides comprehensive experiments covering various non-IID settings and different model architectures, demonstrating the generalizability of the approach.

**Weaknesses:**

- **Trade-off between overhead and performance improvement**: MH-pFedHNGD introduces significant communication and computation overhead compared to MH-pFedHN, but the improvement is marginal. According to *Table 15* and *Table 16*, MH-pFedHN may actually be better than MH-pFedHNGD when considering wall-clock training time rather than just communication rounds. The paper should provide a more thorough analysis of this trade-off and discuss when the distillation variant is worth the additional cost.

- **Poor presentation and unclear writing**: The paper suffers from inconsistent terminology and unclear descriptions that hinder understanding:
  - At *line 292*, the authors define the optimization function of MH-pFedHNG with KL divergence, but at *line 431*, they state "We use MH-pFedHNG to denote MH-pFedHNGD without knowledge distillation," which is confusing and contradictory.
  - In *Figure 2* (step 9), the KL divergence and cross entropy loss labels appear to be reversed.
  - The multi-head design in *Section 3.2* is not clearly explained. It's unclear whether the feature extractor learns representations for each client (mapping $v$ to $f$) and whether the multi-heads ${\varphi_{H_l}}$ are simply multiple MLPs for different model architectures/sizes.

- **Unclear experimental design**: The ablation study in *Section 4.5* lacks clarity regarding the experimental configurations, making it difficult to assess the contribution of different components (see Question 2 below).

**Questions:**

### Q1. Accuracy trends under different *non-IID* settings

For the homogeneous model experiments in *Section 4.2*, *line 359*, the authors mention:
> "Although the accuracy of all methods decreases as the number of clients increases"

However, the results in *Table 1* show that the accuracy actually increases as the number of clients increases for the *non-IID_2* setting (except FedAvg and pFedHN), which contradicts this statement.

- **Could the authors explain why different non-IID settings exhibit different trends?**
- **What are the characteristics of non-IID_2 that lead to this behavior?**
- **Why do FedAvg and pFedHN show a different trend compared with other methods?**

- For the heterogeneous model experiments, most methods achieved the highest accuracy with 100 clients. **Why?**

### Q2. Scalability and participation ratio

The results in *Table 1* and *Table 8* show that the model accuracy of MH-pFedHN and MH-pFedHNGD drops dramatically.

- **Is this mainly because each client has less local data as the number of clients increases?**
- **What if each client has a fixed amount of local data while the number of clients scales (e.g., by sampling a subset of clients to participate in training)?**

- For the experiments with different client participation ratios, **are participating clients selected at the beginning of training or resampled for each communication round?** Could the authors provide the results of both settings?

### Q3. Clarification for the ablation study configuration

In *Section 4.5*, the authors investigate the head components of MH-pFedHN with two configurations, but the description of the second configuration is unclear:
> "(2) MH-pFedHN has one head with an output dimension of N."

Furthermore, at *line 428*, the authors state:
> "When using only a single head, most settings encounter out-of-memory issues, and the rest two cases show limited performance"

This appears to contradict the caption of *Table 3*, which states:
> "Red indicates the setting where each client has one head."

Could the authors clarify:
- Whether in configuration (2), all clients share a single head, or each client has its own dedicated head?
- What are "the rest two cases" mentioned at line 428?
- Can the authors provide results for both the shared head configuration and the dedicated head per client configuration? This would better demonstrate the effectiveness of the multi-head design.

### Q4. Necessity of cross-architecture collaboration

In *Section 4.3* (lines ~386–388), the authors acknowledge a critical issue:

> "Experiment results in Table 2 indicate that methods that allow model heterogeneity show a decline in accuracy under the same settings in most cases for homogeneous models. This suggests that collaboration among heterogeneous clients is still insufficient and challenging, an issue we need to address in the future."

This admission raises a fundamental question about the core contribution of this work. If heterogeneous collaboration leads to accuracy decline compared to homogeneous settings, **what is the practical justification for cross-architecture collaboration beyond merely accommodating resource constraints?**

**Proposed experiment**: Could the authors provide an architecture-specific clustering baseline where clients are grouped by their model architecture (e.g., all ResNet clients train with one hypernetwork, all VGG clients with another, etc.), and compare this with the current mixed-architecture approach?

Additionally, could the authors report the test accuracy for each architecture type separately in the heterogeneous setting (e.g., average accuracy of ResNet clients, VGG clients, MLP clients) and compare them with:
- The same architecture's performance in the homogeneous setting (from Table 1)
- The architecture-specific clustering baseline

This analysis would help clarify:
- Whether certain architectures benefit from cross-architecture collaboration while others suffer
- If there's a trade-off where larger models benefit at the expense of smaller ones (or vice versa)
- Whether the performance decline is inherent to cross-architecture collaboration or a limitation that could be addressed

Without this evidence, it remains unclear whether MH-pFedHN actually enables beneficial "knowledge sharing across different architectures" (as claimed in the abstract and introduction) or whether a simpler architecture-specific clustering approach would achieve better performance while still accommodating resource heterogeneity.

---

### Note · Authors · 2026-01-10

**Comment:**

We would like to withdraw this submission. Thank you to the reviewers for their time and feedback.

**Withdrawal Confirmation:**

I have read and agree with the venue's withdrawal policy on behalf of myself and my co-authors.